# Single reach plans in dorsal premotor cortex during a two-target task

Brian M. Dekleva[1,2], Konrad P. Kording [1,2,3,4,5,6] & Lee E. Miller [1,2,3]

In many situations, we are faced with multiple potential actions, but must wait for more information before knowing which to perform. Movement scientists have long asked whether in these delayed-response situations the brain plans both potential movements simultaneously, or if it simply chooses one and then switches later if necessary. To answer this question, we used simultaneously recorded activity from populations of neurons in macaque dorsal premotor cortex to track moment-by-moment deliberation between two potential reach targets. We found that the neural activity only ever indicated a single-reach plan (with some targets favored more than others), and that initial plans often predicted the monkeys' behavior on both Free-Choice trials and incorrect Cued trials. Our results suggest that premotor cortex plans only one option at a time, and that decisions are strongly influenced by the initial response to the available set of movement options.

[1] Department of Biomedical Engineering, Northwestern University, 633 Clark St, Evanston, Chicago, IL 60208, USA. [2] Department of Physiology, Northwestern University, 303 E Chicago Ave, M211, Chicago, IL 60611, USA. [3] Department of Physical Medicine and Rehabilitation, Northwestern University, Chicago, IL 60611, USA. [4] Department of Applied Mathematics, Northwestern University, Chicago, IL 60208, USA. [5] Department of Neuroscience, University of Pennsylvania, Philadelphia, IL 19104, USA. [6] Department of Bioengineering, University of Pennsylvania, 210S 33rd St, Suite 240 Skirkanich Hall, Philadelphia, IL 19104, USA. Correspondence and requests for materials should be addressed to L.E.M. (email: lm@northwestern.edu)

We often prepare for a movement by surveying the available actions, then waiting for more information before moving. For example, a tennis player waiting to receive a serve can anticipate the need to perform a forehand or backhand return but can observe the trajectory of the ball before deciding between those two options. Likewise, we can plan to grasp an object without yet knowing the most useful hand posture. Similar parallels can be drawn for almost all types of movements, indicating that deliberation between potential actions is a ubiquitous aspect of motor control.

Although the initial deliberation between multiple potential actions is a widespread phenomenon, the neural processes underlying it are largely unknown. The brain could adopt two general approaches: (1) simultaneously represent several potential options; or (2) initially represent only a single plan (perhaps a compromise), then later switch if necessary. Several behavioral studies have attempted to disentangle these possibilities, most notably through the go-before-you-know paradigm in which a subject is given multiple reach targets and then forced to move before knowing which is correct. Early movement trajectories are often directed in between the two options, which some have interpreted as a spatial averaging of two simultaneous plans[1–6]. However, an intermediate movement is not necessarily indicative of multiple simultaneous plans and might instead reflect a single plan optimized for the uncertain task[7,8]. Thus, the neural processes that underlie the deliberation between potential movements cannot be readily interpreted from the movements themselves.

A few studies have used individual neurons recorded in motor cortex to probe planning-related activity in the face of multiple discrete movement options, with results that support the "dual representation" hypothesis[9–13]. However, these studies relied on single-electrode recordings, using trial-averaged data to reduce noise, with the implicit assumption that all trials reflect a single consistent process. Since delay-period planning activity has no measurable external signature, it is unclear how well this assumption actually holds. To examine the wholly internal neural processes at play during movement deliberation requires analysis on the timescale of individual trials.

An alternative to trial averaging is to combine the information obtained from many simultaneously recorded neurons. This population-based approach has increasingly been adopted in motor areas, where it provides a reliable estimate of limb movement[14–21]. An important, less-exploited advantage of simultaneous recordings is the ability to probe neural processes that have no measurable behavioral outcome. Several studies have used population recordings from cortex to identify changes of mind on single trials of a multiple-potential-target-reaching task[22,23]. This ability to interpret activity on a short timescale in the absence of behavioral correlates provides a possible means to investigate deliberation between initial movement options.

Here we used simultaneous recordings from dorsal premotor cortex (PMd) of the macaque monkey to monitor the development of movement plans in the face of two potential movement options. Using dimensionality reduction methods to extract underlying low-dimensional latent signals from the population activity[14–21,24] we tracked the moment-by-moment planning in these areas during single trials. We found that when presented with two opposing reach options, PMd quickly developed a movement plan for only one, planning some targets more often than others. The initial plans were strongly predictive of various aspects of the monkey's eventual behavior (including target choices when the correct target was not specified, reaction times, and task errors). However, our results showed no any evidence of simultaneous representation in premotor cortex during the deliberation between multiple options. Instead they indicate that

ultimate movement decisions are strongly influenced by an initial reach plan corresponding to one of the potential choices.

## Results

**Experimental setup and trial-averaged neural responses**. We trained two rhesus macaques on a center-out reaching task in which they controlled an on-screen cursor using a planar manipulandum (Fig. 1a). Each session consisted of both 1-Target and 2-Target trials, randomly interspersed. On 2-Target trials the monkey first positioned the cursor in the central target, after which two additional targets appeared located 180° apart with respect to the center (Fig. 1c, left). These targets remained on screen for between 750 and 1000 ms (Target On), then disappeared for 250–500 ms (Target Blank) before a single target reappeared for 250–500 ms (Cue). Finally, the central target disappeared, the second outer target reappeared, and a tone cued the monkey to reach to the originally cued target (Go). 1-Target trials followed the same basic structure, except that the monkey was only ever shown one outer target (Fig. 1d, left). On roughly 10% of 2-Target trials (Free-Choice trials) we omitted the Cue epoch, and instead instructed the monkey to reach immediately after the Target Blank epoch (Fig. 1c, bottom). In all conditions, the monkeys made approximately straight reaches to the targets (Fig. 1c, d, right).

Throughout each session, we recorded the activity of discriminated neurons in both PMd and primary motor cortex (M1) using chronically implanted 96-electrode arrays (Blackrock microsystems). We calculated firing rates by filtering the spike trains with a causal, half-Gaussian kernel before downsampling to 40 Hz. Most PMd neurons appeared directionally tuned throughout the planning phases (i.e., Target On, Target Blank, and Cue epochs) of the 1-Target task. To summarize the population-wide responses, we calculated a preferred direction for each neuron using a standard cosine model[25]. We then used those preferred directions to categorize the response of each neuron on every trial as either pro-PD (target at preferred direction), anti-PD (target opposite preferred direction), or orthogonal-PD (target orthogonal to preferred direction). The traces in Fig. 2a show the average of all pro-PD and anti-PD activity across 1-Target trials after subtracting orthogonal-PD activity to account for changes not due to reach direction. The resulting plot reveals an increase in pro-PD activity shortly after target presentation and a decrease in anti-PD activity, corroborating previous findings of sustained, planning-related activity in PMd during reaching[26–30]. We repeated this procedure for 2-Target trials and found an increase in both pro-PD and anti-PD activity (Fig. 2b), seeming to indicate dual representation of both targets, as previously reported[9–13].

**Single-neuron tests of dual representation**. We further tested the nature of the apparent dual representation effect across 2-Target trials using the control analysis first described by Cisek and Kalaska[11]. Briefly, on each trial we labeled the planning-related activity as described above (pro-PD, anti-PD, and orthogonal-PD), and calculated the percentage of pro-PD and anti-PD activity that exceeded the median orthogonal-PD activity. Since this metric grouped pro-PD and anti-PD activity together, a dataset consisting only of single-target representations should have resulted in values around 50% (pro-PD activity would exceed the orthogonal-PD median threshold but anti-PD activity would not). Indeed, the black histogram in Fig. 2c—which represents the activity during 1-Target trials—shows that the proportion of high pro-PD and anti-PD activity across neurons was not significantly different from 50% (t-test, $p = 0.1$). In contrast, the distribution of high activity on 2-Target trials (Fig. 2c, blue) was skewed significantly toward 100%

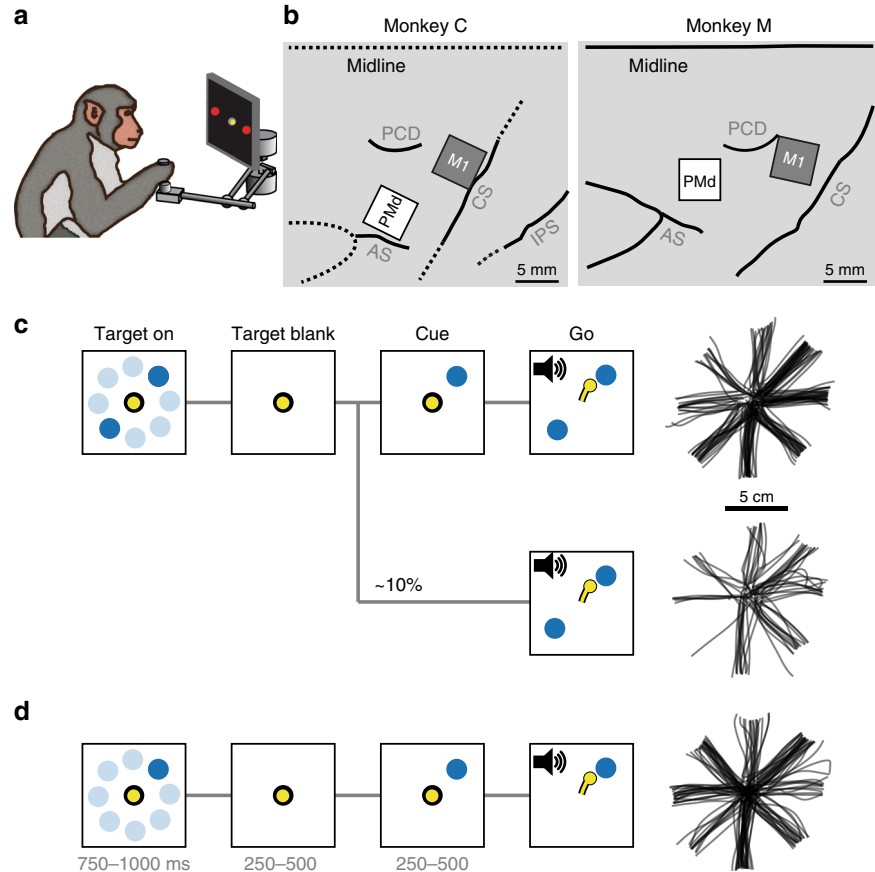

**Fig. 1** Experimental setup. **a** Monkeys used a planar manipulandum to control a cursor. **b** Array placement in dorsal premotor (PMd) and primary motor (M1) cortex (CS central sulcus, AS arcuate sulcus, PCD precentral dimple, IPS intraparietal sulcus). Dashed lines correspond to approximate locations of sulcal landmarks that were obscured by the boundaries of the intraoperative craniotomy. **c** Task events during 2-Target trials and resulting cursor trajectories. The monkeys were initially presented with two opposing targets. On most trials (top), the correct target was displayed at the start of the Cue epoch; roughly 10% did not contain a Cue epoch (Free-Choice, bottom). On these trials, the monkeys were forced to select one of the two targets with no information about which would lead to reward. **d** Same as in **c** for 1-Target trials. The monkey image in **a** was created by Paul Wanda for use in this paper. All rights reserved

($t$-test, $p < 10^{-9}$), consistent with activity related to both targets—the result expected of a dual representation paradigm.

Although the results from Fig. 2a–c seem to provide convincing evidence of dual representation, we considered the possibility that they were instead caused by target biases during the planning phase. The analysis in Fig. 2c implicitly assumes a null hypothesis of unbiased guessing, in which the subject randomly selects one of the two displayed targets at the start of each trial. However, there is another possible guessing-based approach, which we term biased guessing. Under a biased guessing paradigm, the subject still only ever creates an initial plan to one target, but that plan is almost always to the same target. For example, for every 2-Target trial with a left and right target, the subject might start by planning a reach to the right (switching later if cued to the left target). We simulated datasets for both of these guessing-based approaches (see Methods) to determine whether the analysis from Fig. 2c could correctly identify that they contained only single-reach plans. For the unbiased guessing dataset, the analysis returned a distribution of high pro-PD and anti-PD activity that was not significantly different from 50% (Fig. 2d; $t$-test, $p = 0.13$), correctly ruling out the possibility of dual representation. However, for the biased guessing dataset, the analysis returned a highly right-skewed distribution (Fig. 2e; $t$-test, $p < 10^{-10}$). This result strongly—but incorrectly—suggests the presence of dual representation. Thus, while the analysis from Fig. 2c can discount the possibility of an

unbiased guess-and-switch approach, it cannot rule out the possibility that the monkeys employed a single-plan strategy with strong biases in the initial plans.

**Population-based approach to single-trial reach plan decoding.** The conflicting interpretations in Fig. 2 indicate that methods based on the trial-averaged responses of single neurons can lead to incorrect conclusions about the nature of neural responses on individual trials. To disambiguate the initial target deliberation requires that planning-related neural responses be evaluated at the level of single trials. Array recordings allow the large number of recorded neurons, rather than a large number of trials, be used to compensate for the stochastic neural noise. As a first step, we used principal component analysis (PCA) to reduce the population's activity to a de-noised, 10-dimensional neural state[14,16,22,31–34]. Figure 3a shows the evolution of three dimensions of this state for left and right 1-Target trials, superimposed on the neural states observed across all left and right 1-Target trials. Even in three dimensions, the reach directions are separable, indicating that a low-dimensionality-based approach can provide a useful readout of instantaneous reach planning.

To quantify the extent of target-related activity within the neural state space, we developed a proximity metric based on the Mahalanobis distance to clusters associated with each reach direction (see Methods). As an example, take a point in the

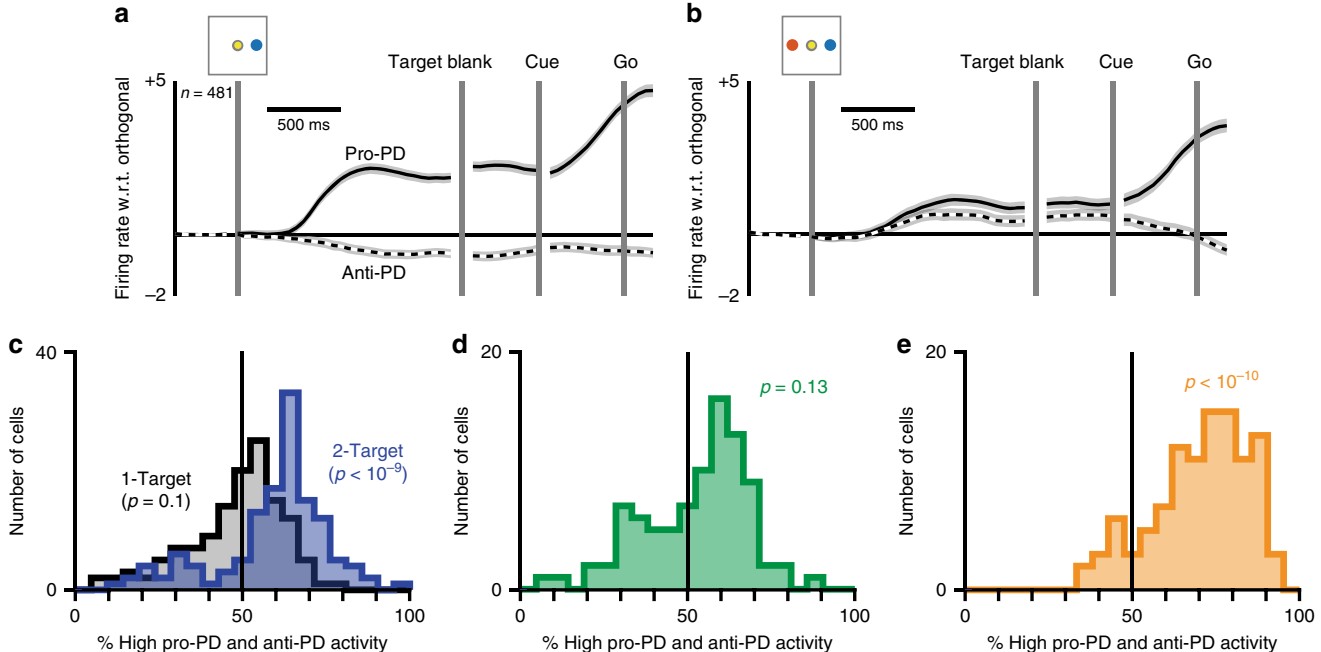

**Fig. 2** Apparent dual representation from trial averaging. **a** Average activity traces of neurons with preferred directions aligned (solid) and anti-aligned (dashed) with the reach direction across all 1-Target trials. Activity of neurons with preferred directions orthogonal to cue direction has been subtracted. Shading represents bootstrapped 95% confidence. **b** Average traces (as in **a**) for all 2-Target trials. **c** Histograms showing the percentage of activity during pro-PD and anti-PD trials that exceeded the median activity observed on orthogonal-PD trials. Rightward skew on 2-Target trials (blue) suggests simultaneous representation. p-Values correspond to the null hypothesis that the distribution mean is 50% (t-test). **d** Analysis as in **c** for a simulation of an unbiased guessing approach using 1-Target data. **e** Analysis as in **c** for a simulation of a biased guessing approach using 1-Target data

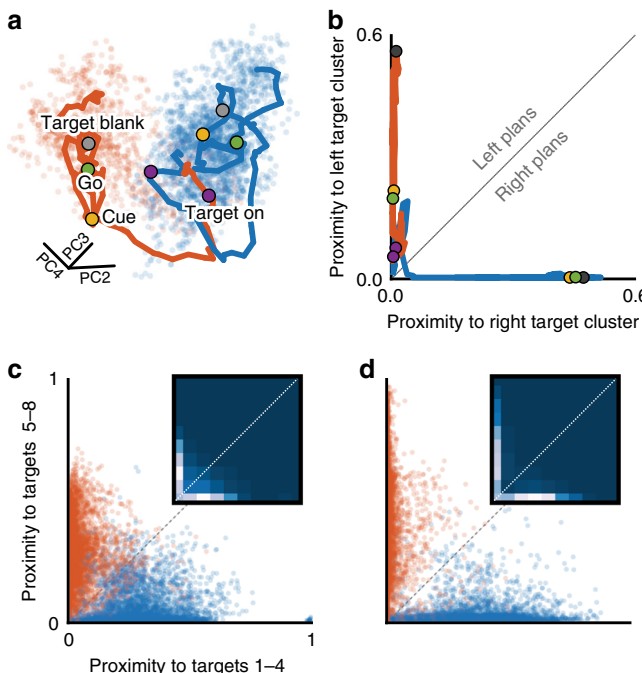

**Fig. 3** Low-dimensional neural states on 1-Target trials. **a** Low-dimensional neural state trajectories throughout individual left (red) and right (blue) 1-Target trials. Blue and red clusters reflect all neural states visited during 1-Target trials to rightward and leftward targets, respectively. Colored, labeled markers indicate the first time points in each trial epoch. **b** Proximity plot (x-axis: proximity to the rightward cluster; y-axis: proximity to leftward cluster) for the example trials from **a**, with corresponding trial event markers.
**c** Proximity plot of Target Blank activity over all 1-Target trials, monkey C. Inset density plot shows that most points lie along the x- and y-axes.
**d** Same as in **c** for monkey M

leftward trial trace (red) from Fig. 3a. We can calculate its proximity to the leftward reach cluster (red point cloud) as well as its proximity to the rightward reach cluster (blue point cloud). Plotting these two values against each other for all points along the example leftward reach neural state trajectory results in the red trace in Fig. 3b. Repeating for the rightward trial in Fig. 3a gives rise to the blue trace in Fig. 3b. Excursions along the y- and x-axes can be interpreted as evolving leftward and rightward reach plans, respectively. To display the results from multiple target axes, we arbitrarily chose the x-axis to represent half of the targets (at 0°, 45°, 90°, and 135°) and the y-axis to the other half. Over all sessions and reach directions, we successfully classified the final reach direction from neural data collected during the Target Blank epoch of 1-Target trials for both monkey C (Fig. 3c, 97%) and monkey M (Fig. 3d, 98%).

**Neural state test of dual representation.** Since the state space proximity metric proved capable of accurately describing population-wide planning on 1-Target trials, we next applied it to activity on 2-Target trials. To ensure that the method could differentiate between different types of planning, we first simulated three potential models: dual-target; averaged-plan; and stay-or-switch. The dual-target and averaged-plan models reflect mechanisms by which PMd could represent two reach directions simultaneously. Alternatively, the stay-or-switch model represents the case in which PMd plans only a single reach at a given time. For all models, we simulated activity only during the Target Blank epoch.

To model dual-target representation, we first fit cosine tuning curves to 1-Target data for all neurons. We transformed those single-target tuning curves into bimodal, dual representation curves to match the response types reported by Cisek and Kalaska[11]. Figure 4a shows a schematic of averaged responses (as in Fig. 2) and representative 1-Target (unimodal) and 2-Target

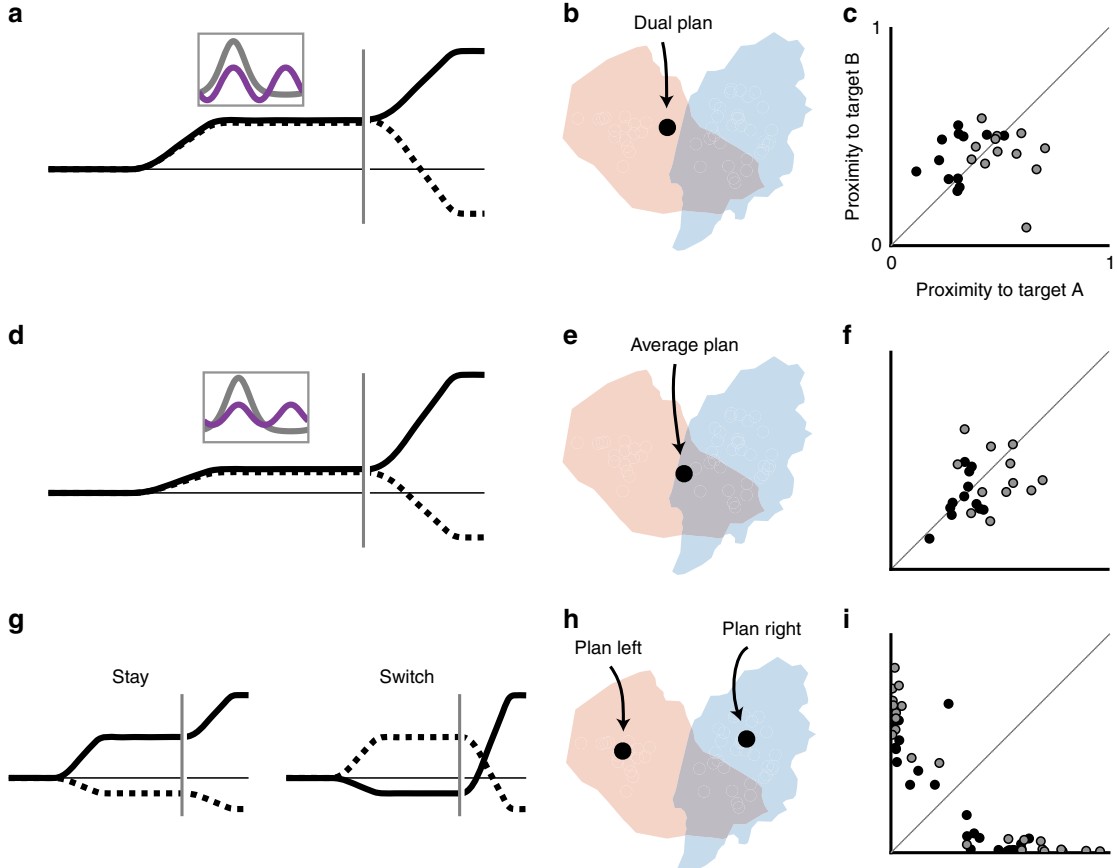

**Fig. 4** Model predictions of low-dimensional states. **a** Illustration of pro-PD (solid) and anti-PD (dashed) activity on a 2-Target trial under a dual-target model based on Cisek and Kalaska[11]. Inset shows example 1-Target (gray) and 2-Target (purple) turning curves. **b** Simulated activity from the model in **a** for a left/right trial results in a low-dimensional neural state between the left and right 1-Target planning states. **c** Proximity plot of the simulated simultaneous representation states for all target axes. Symbol color corresponds to monkey (black, monkey C; gray, monkey M). **d–f** Same for averaged-plan model. **g–i** Same for stay-or-switch model

(bimodal) tuning curves. From the bimodal curves, we simulated activity for the left/right target pair from Fig. 3a; the resulting neural state was close to both the left- and right-plan clusters (Fig. 4b). Across all target pairs and sessions, simulated dual-target neural states tended to lie near both reach direction clusters (Fig. 4c, points close to diagonal).

The averaged-plan model represented another variant of dual representation; each neuron's response reflected the average of its responses to each target individually. Its simulated activity was qualitatively similar to the dual-target model (Fig. 4d–f), with neural states near both clusters.

Neural responses for the third model (stay-or-switch) corresponded to only one reach plan at any given time (Fig. 4g); the associated neural states (Fig. 4h) and proximity plots (Fig. 4i) were indistinguishable from the 1-Target condition. The clear difference in results between simulated dual- and single-representation models thus demonstrated the ability of our population-based proximity metric to identify dual representation in single-trial neural activity.

In Fig. 5a, we show the neural states observed during two epochs—Target Blank and Go—across all left/right 2-Target trials from the example session in Fig. 3a. At the time of movement (Fig. 5a, right plot) the neural states clearly reflected the cued direction. However, during the earlier Target Blank epoch (Fig. 5a, left plot), they appeared to indicate a rightward reach plan. The proximity plots incorporating all 10 dimensions of the neural state confirm the appearance of the two-dimensional projection: pre-Cue activity almost always reflected a rightward reach

(Fig. 5b, left—high density along $x$- and $y$-axes). Across all targets and sessions, the neural states during the Go epoch accurately reflected the cued reach directions (Fig. 5c, d, right plots). However, during the Target Blank epoch, the proximity plots suggest a mix of cued and anti-cued reach plans. This is expected, since the monkeys did not yet know which target was correct. Of note, the proximity plots reveal excursions mainly along the axes, as in 1-Target trials. In the scatter plots, this effect is more apparent for monkey M (Fig. 5d, left), but density plot insets reveal that both monkeys exhibited what appears to be largely single-target planning activity.

To quantify the extent to which the observed neural states represented single or dual reach plans, we developed a new metric from the calculated proximities, called the dual representation index (DRI; see Methods). Figure 6a shows the 1-Target proximity plot from Fig. 3c overlaid with a contour plot of the DRI metric. The DRI is bounded by zero (along either axis) and one (upper right corner), with highest values nearest the diagonal (indicative of dual representation; Fig. 4c, f). We first calculated the DRI for neural states observed during the Target Blank epoch on 1-Target trials. The resulting histogram for monkey C is shown in Fig. 6b. DRI < 0.2 captured all but 3% of 1-Target planning for monkey C, and all but 1% for monkey M. We then calculated DRI for both actual (Fig. 6e, f; black) and simulated 2-Target neural states (Fig. 6e, f; gray). While DRI for the two simulated models largely exceeded 0.2 (84%, monkey C; 87%, monkey M), the percentage of actual 2-Target activity above the threshold was very low (6%, monkey C; 3%, monkey M). Thus,

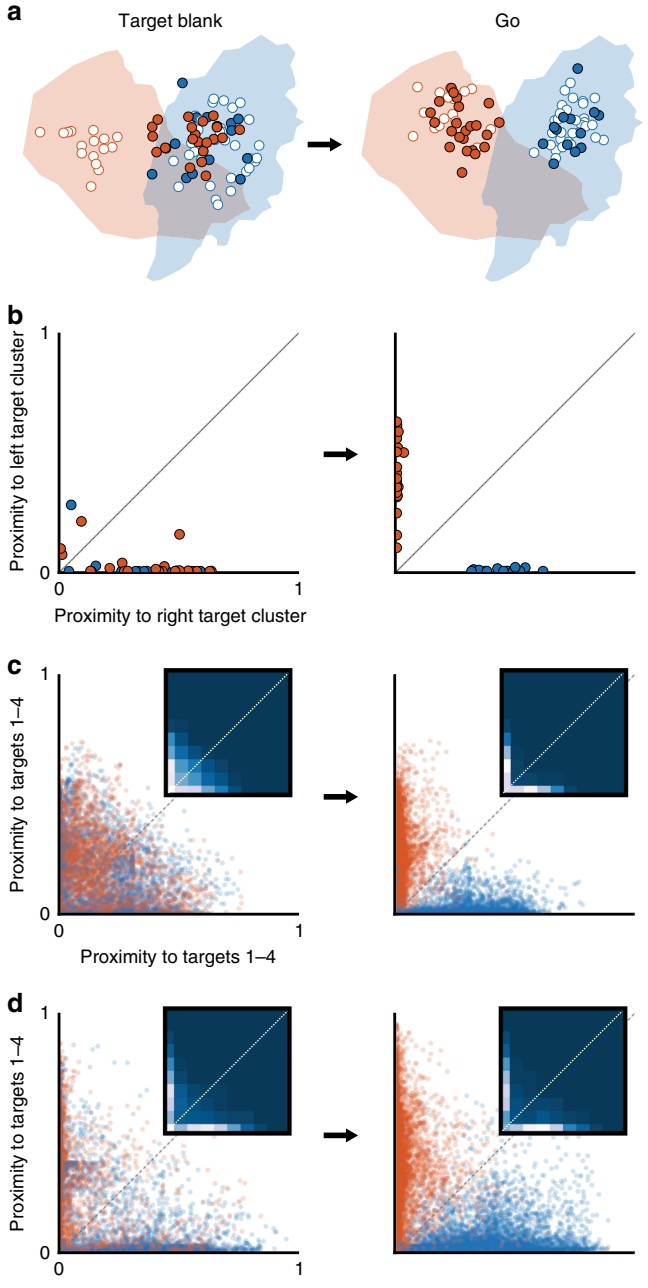

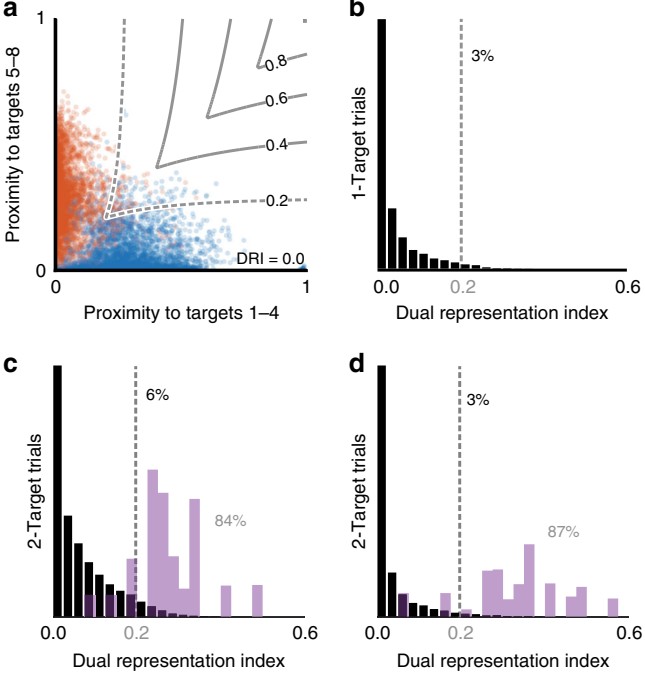

**Fig. 6** Testing for dual representation in low-dimensional states. **a** Proximity plot of 1-Target trials for monkey C from Fig. 3c, overlaid with a contour plot of the dual representation index (DRI). **b** Histogram of DRI values from **a**. A post hoc upper threshold of DRI = 0.2 (vertical dashed line) captured all but 3% of 1-Target activity for monkey C and all but 1% for monkey M. **c** Histogram of DRI values for all 2-Target trials (black), monkey C. Purple bars reflect simulations from both the dual-target model and the averaged-plan model. **d** Same as **c** for monkey M

**Fig. 5** Low-dimensional neural states on 2-Target trials. **a** Neural states during planning (left plot) and movement execution (right plot) for all left/right target trials during an example session. Filled red circles correspond to trials with a cue to the left target, filled blue circles a cue to the right target. Open circles represent 1-Target left (red) and right (blue) trials. **b** Proximity plots of the left/right 2-Target trials in **a**. **c** Left: proximity plots of planning activity (Target Blank epoch) for all 2-Target trials, monkey C. Inset heat map shows the high concentration of points along the x- and y-axes. Right: proximity plots of execution-related activity (Go epoch) for all 2-Target trials, monkey C. **d** Same as **c** for monkey M

although the trial-averaged results from Fig. 2 suggested the presence of dual representation on 2-Target trials, we found no such evidence in single-trial population activity. This result held even if we restricted our analysis to only the subpopulation of neurons with trial-averaged responses most strongly indicative of dual representation (see Supplementary Note 2 and Supplementary Figure 2).

**Target preferences during 2-Target planning**. The results from the above analyses suggest that even on 2-Target trials, the monkeys planned for only one reach at a time. To track the evolution of these plans, we calculated the difference over time between the proximities to the pair of relevant reach direction clusters. The resulting difference (ΔProximity) provided an estimate of the direction and strength of the instantaneous reach plan. This metric was more conservative than a probability-based classifier in that it was less likely to report spurious reach plans having low strength (see Supplementary Figures 3 and 4). Figure 7a shows example single-trial target decodes using this metric for left/right, 2-Target trials. At the time of movement execution (Go epoch) reaches to the left were clearly distinguishable from those to the right. However, activity early in the trials overwhelmingly (88%) resembled rightward reach plans. Both monkeys had similarly skewed preferences (Fig. 7b). However, these preferences were inconsistent across sessions and did not always follow a clear hemispatial bias. On the first session, monkey M had an up/right preference. Over the next two sessions, those preferences appeared to rotate to the downward and leftward directions (Fig. 7c, bottom row). Monkey C had similar preferences during the first two sessions, but completely opposite preferences for two of the four target pairs on the third session (Fig. 7c, top row). The inconsistency of the planning preferences, both within and across subjects, suggests that they arose as part of conscious task strategies rather than from an ingrained predilection for minimizing exertion or some other physiological cost function[35–37].

We considered the possibility that the decoded biases during early planning periods resulted from anisometries in the proximity metric across reach directions, and thus did not accurately reflect

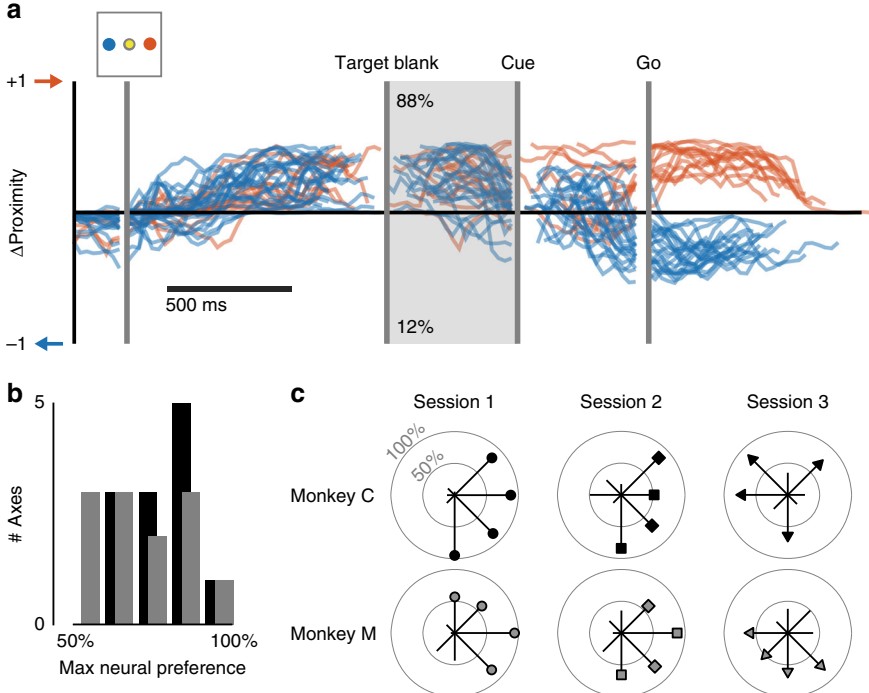

**Fig. 7** Preferential reach planning in PMd during two-target trials. **a** ΔProximity traces for all left/right trials in an example session (monkey C, session 1). Blue traces indicate those for which the monkey was cued to the left. Red traces indicate trials cued to the right. During the Target Blank epoch, 88% of activity indicated a rightward reach plan. Activity prior to Target Blank epoch is aligned to Target On. Activity during the Target Blank epoch is aligned to Cue appearance. Activity after Cue appearance is aligned to the Go cue. **b** Distribution of neural preferences across target axes from all sessions (black, monkey C; gray, monkey M). **c** Neural preferences across sessions. For each axis, radial length represents the observed preference

the instantaneous plan. To address this, we compared each monkey's decoded target preferences on cued 2-Target trials to their actual choice biases on Free-Choice trials. Over all 16 left/right Free-Choice trials from the session in Fig. 7, monkey C reached to the right target 11 times (69%, Fig. 8a). This rightward choice bias mirrored the preference observed in the initial planning activity on cued 2-Target trials (88%, Fig. 7a). Across monkeys, we found a strong correspondence between the target preferences decoded from cued 2-Target neural activity and the actual reach direction on Free-Choice trials (Fig. 8b). Additionally, reach plans decoded from primary motor cortex (M1) matched those from PMd (see Supplementary Note 1 and Supplementary Figure 1) further indicating that the neural state decoding accurately reflected the monkeys' instantaneous reach plans.

On average, the monkeys had a 75:25% target preference during planning (Fig. 7b). Thus, at times, they began planning to move to a target they did not generally favor. While the results from Fig. 8b show that preferences in early planning activity were largely predictive of Free-Choice movement preferences, it was unclear how well the planning activity on a single Free-Choice trial might correlate with the eventual movement decision. Were initial plans made to favored targets more likely to be carried through to execution than those made to non-favored targets? To answer this question, we identified the favored targets during the Target Blank epoch on 2-Target trials. We then compared the direction decoded during Target Blank on each Free-Choice trial with the eventual reach direction. We found no difference between trials with initial plans to favored targets compared to non-favored targets; neural activity for both was equally predictive (Fig. 8c). Thus, while the monkeys did exhibit clear session-wide target preferences, it was the initial plan on any given Free-Choice trial that best predicted the chosen reach direction.

**Plan strength and reaction time.** Our findings suggest the existence of only single-reach plans, but the strengths of those plans varied widely across trials. Consider the left/right responses in Fig. 7a. Most early trial activity clearly indicated rightward reach plans. However, some activity suggested leftward (negative ΔProximity) or weak-to-nonexistent plans (ΔProximity near zero). Due to the conservative nature of the ΔProximity metric (see Supplementary Note 3 and Supplementary Figures 3 and 4), large magnitudes—either positive or negative—could only arise from neural states close to a given reach direction cluster. Due to the lack of evidence for dual representation in our data (Fig. 6), low magnitudes could only arise from neural states unassociated with any specific reach plan. We predicted that the magnitudes of the decoded reach plans on individual trials would correlate with some aspect of the kinematics of the executed movement. To test this, we compared the average magnitude of ΔProximity in a 100 ms window preceding the Go epoch on each cued, 2-Target trial to the subsequent movement reaction time. Figure 9a shows this relationship for left-cued reaches on an example session. Reaction times were markedly shorter when ΔProximity indicated a strong leftward plan at the time of the Go cue, and longer when it indicated a rightward plan. This negative correlation between decoded plan strength to the cued target and reaction time occurred for both monkeys (Fig. 9b; linear mixed effects model accounting for differences across reach directions and sessions; monkey C: coeff = −127, $F$-test $p = 0.0036$; monkey M: coeff = −159, $F$-test $p \approx 0$) and is similar to observations in previous studies[12,31,38]. We found a similar negative correlation between ΔProximity directly preceding the Go cue and reaction time on Free-Choice trials (Fig. 9c, d; monkey C: coeff = −241, $F$-test $p = 0.0017$; monkey M: coeff = −108, $F$-test $p = 0.0266$).

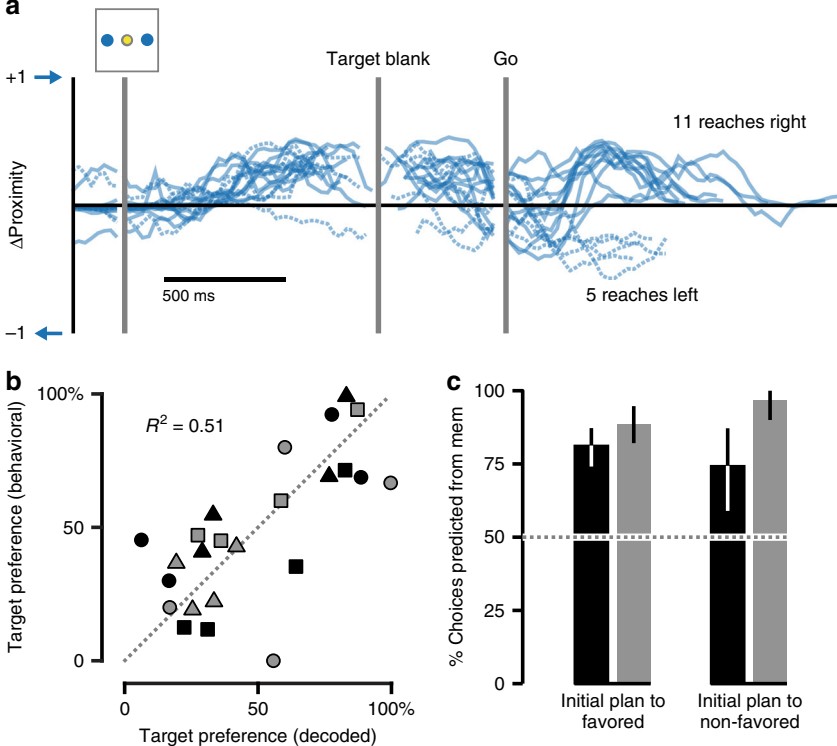

**Fig. 8** Neural preferences match Free-Choice behavior. **a** ΔProximity traces for all left/right Free-Choice trials in the example session from Fig. 4a. The monkey chose the rightward target on 11 of the 16 trials. **b** Across all sessions, the monkeys' decisions on Free-Choice trials for each axis correlated with their planning preferences decoded from cued 2-Target trials. Black symbols correspond to monkey C, gray symbols to monkey M. Symbol shape corresponds to the different sessions, as in Fig. 7c. **c** Bar plot indicating the degree to which Target Blank period activity successfully predicted the monkeys' choices on Free-Choice trials for each session and target axis. Black bars correspond to monkey C, gray to monkey M. Error bars represent 95% confidence bounds calculated from the standard error

**Reach planning on error trials**. Although both monkeys understood the 2-Target task, at times they chose the incorrect—i.e., non-cued—target (monkey C: 16%, monkey M: 27%). We examined the progression of reach plans during these error trials to reveal potential sources of incorrect movement choices. Using the reach plans decoded early (Target Blank) and late (Cue) in the trial, we characterized three types of errors, which together described over 97% of all errors. During type 1 errors, the monkey maintained a consistent reach plan throughout, not deviating even after receiving a cue to the opposite target (Fig. 10a, heavy trace). During type 2 errors, the monkey switched to the correct plan after cue presentation, but then later reverted to his initial incorrect plan upon receiving the Go cue (Fig. 10b, heavy trace). During type 3 errors, the monkey switched to the incorrect reach plan at the last second, despite having planned correctly throughout the trial (Fig. 10c, heavy trace). Despite different error rates between monkeys, the high incidence of type 1 and 2 errors suggests that both monkeys had an aversion to switching away from their initial reach plans. This was true whether that initial plan was to a favored target or to a non-favored target (Fig. 10d; filled vs. open bars).

**Discussion**

We found that when faced with two potential reach targets, activity in PMd represented only a single movement plan despite several previous studies supporting a dual representation mechanism[3,4,10–13,39–41]. We cannot discount the possibility the monkeys in our study adopted a different strategy than those in previous studies. Our task, unlike others, did not require any memorization and enforced only lax time constraints. This

simplicity in design may have led to a similarly simplistic (i.e., single-target) planning approach. However, it may be that premotor cortex is incapable of planning multiple movements simultaneously, and that the trial averaging used in previous studies blurred the distinction between dual representations and biased single-target representations (see Supplementary Discussion). When we replicated the trial-averaged analysis from Cisek and Kalaska meant to control for a guess-and-switch strategy[11], our results also appeared to support multiple encoding (Fig. 2c). However, additional simulations showed that this finding most likely resulted from strong preferences for specific targets (Figs. 2e and 7).

It could be that dual representations develop in PMd only after extensive training on a two-target task. Work in the field of brain-computer interfaces has shown that motor cortex can generate novel patterns of activity after learning[21,42–46]. We therefore cannot claim that PMd is incapable of responses more complex than those we observed. However, if dual reach representation in PMd arises only as a result of overlearning, it might not be a typical or natural component of decision-making. Additional studies of PMd throughout long-term learning might clarify this possibility.

By its very nature, the concept of dual reach representation invokes a representational view of cortex. That is, it assumes that firing rates reflect the explicit encoding of particular movement features (e.g., reach direction). Recent work has challenged this assumption, suggesting instead that motor and premotor cortices operate as a dynamical system, with much of its activity reflecting these dynamical properties[14,33,47]. This view proposes that PMd activity defines the initial goal-related state from which activity evolves upon movement initiation. In this framework, perhaps

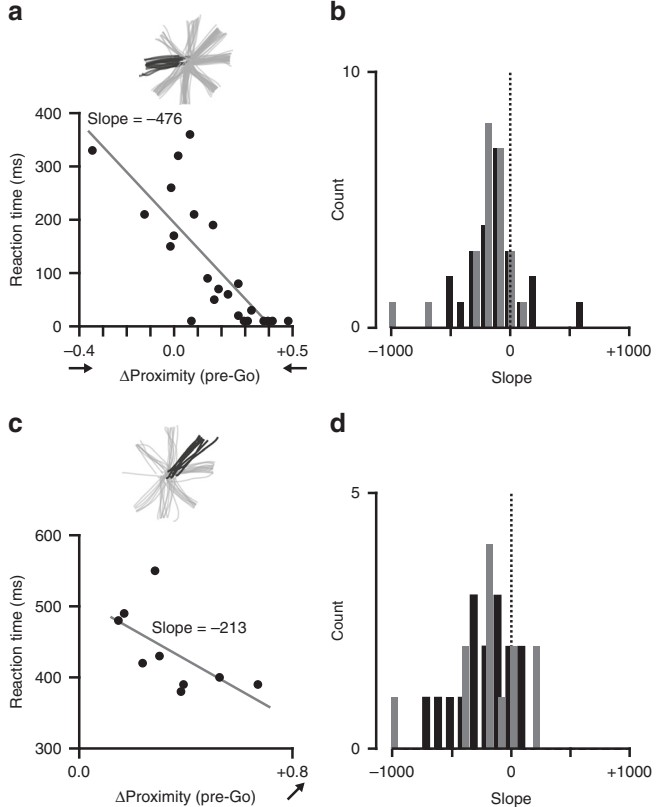

**Fig. 9** Correlation between pre-Go neural state and reaction time.
**a** Reaction time as a function of the strength of the reach representation in PMd calculated 100 ms prior to the Go cue for all leftward reaches on two-target trials during a single session. Reaction times decreased as the decoded strength of the neural representation increased. **b** Histogram of the slopes (as calculated in **a**) for all target axes from all sessions (monkey C, black; monkey M, gray). **c** As in **a** for all reaches to the upper left target during Free-Choice trials. Reaction time decreased with the strength of the decoded representation. **d** As in **b** for Free-Choice trials

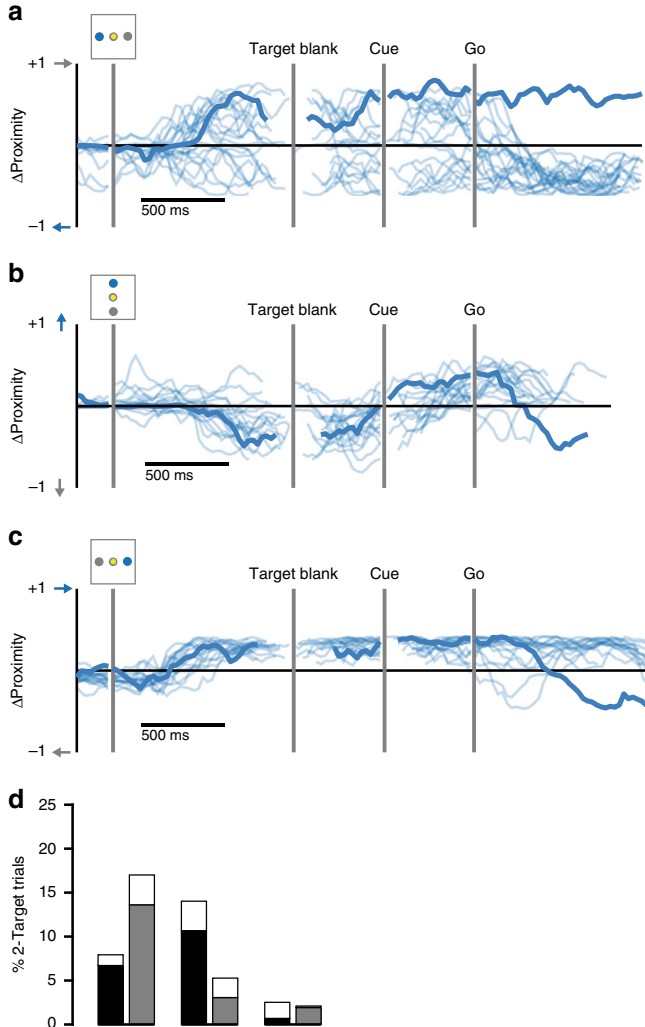

**Fig. 10** Heterogeneity in types of errors. **a** An example error (thick line) made to the rightward target when cued to the left (monkey M, session 2). This type of error can be characterized as an early plan that is not changed even after receiving a contradictory cue (type 1). Light traces indicate all correct trials with the same leftward cue. **b** An incorrect reach to the bottom target (monkey C, session 1). This type of error can be characterized as a late reversion to the initial plan (type 2). **c** An incorrect reach to the leftward target (monkey M, session 1). This error represents a late switch away from the correct reach plan (type 3). **d** Histogram showing the frequency of each error type for both monkeys (monkey C, black; monkey M, gray). Filled and open segments indicate whether the errors were made to a favored or non-favored target, respectively

PMd could achieve an intermediary neural state when faced with multiple movement options. However, it is not clear that this new neural state could be considered dual representation in the traditional sense. Instead, it may simply reflect a movement plan that can be easily switched to either of the presented targets, an interpretation supported by recent behavioral results[7,8]. Additional work is needed to interpret the neural responses occurring during complex decision-making tasks and to test the representational and dynamical systems hypotheses more directly.

The results from this study highlight the potential influence of preferences and biases on trial-averaged analyses of planning-phase activity. This concern cannot be summarily dismissed, as directional biases appear to be quite common in two-target tasks[10,22,48]. Some studies have attempted to minimize the effect of preferences by explicitly adjusting reward values[10] or other aspects of the task[22] to encourage reaches to non-favored targets. Carefully executed, these approaches can successfully equalize the number of movements to each target. However, eliminating biases in the distribution of final reach directions does not necessarily eliminate biases during planning phases. We found that while early planning activity largely predicted Free-Choice decisions, it was not without exception. There were some trials for which PMd activity suggested one reach direction throughout the entire trial, only to switch moments before movement execution (Fig. 10c). This last-second switching was infrequent, but a task

designed to balance choices across all directions may encourage it. If so, analysis based on the assumption that a subject has considered all targets equally throughout the entirety of the trial may not be valid. Trial-averaging methods simply cannot provide the temporal resolution necessary for determining instantaneous, potentially transient states of mind.

Although the monkeys in our study displayed quite strong preferences, the source of those preferences is not obvious. It seems that they reflected the monkeys' attempts at strategies that they believed might increase the likelihood of reward. Since the reward structure on Free-Choice trials was completely random, no such successful strategy was possible. The apparent randomness of the target preferences may therefore reflect the monkeys'

guesses and over-interpretation of brief patterns in the target presentations. Determining the source of target preferences and their effect on planning would require further experiments with different reward structures.

The degree to which the initial reach plan predicted Free-Choice behavior suggests that early responses in PMd strongly influence the eventual movement decision. Supporting this idea is the observation that early reach plans on error trials almost always matched the movement direction (type 1 and type 2 errors accounted for over 90% of all errors; Fig. 10d). That is, the monkeys made errors either because they unwaveringly stuck with their initial plan or switched back to it after briefly considering the other (correct) option. Changes of mind have been observed before in cortex[22,23,44–51], but not in direct contradiction to an explicit visual cue. In these cases, it appears that the monkey's first reaction to the pair of targets carried more weight than the subsequent—and completely informative—visual cue. This might indicate influence from another brain area overriding sensory inputs, or biases within PMd that resist changes away from an initial plan.

Recording from a large neural population allowed us to observe planning-related activity on a short timescale. This revealed a strong relationship between the instantaneous neural state at the time of the Go cue and the reaction time of the subsequent movement. Trials with strong plans to the correct target often had very short reaction times (Fig. 9a), with some reaches even beginning in anticipation of the Go cue[52,53]. Alternatively, trials with incorrect plans—even strongly so—at the time of the Go cue often displayed long reaction times, indicating a time cost associated with switching the plan. The ability to hasten or delay movement initiation upon evaluation of the current movement plan (and determining whether a change must be made) suggests that properties of the plan alone cannot predict reaction time. Instead, our results argue for a more complex mechanism governing the transition from movement preparation to movement execution, and support the view that movement planning and initiation may well involve separate neural processes[54].

Trial-averaged analysis methods have contributed a great deal to our understanding of movement planning and decision-making. However, at times they may lead to an oversimplified view of neural processes. The ability to interpret a quickly changing neural state with high temporal resolution is essential when studying high-order brain functions like decision-making. The heterogeneity in single-trial responses observed in this study and others suggests that decision-making cannot be fully explained by models that assume highly predictable neural responses to a given task condition. Large-scale recordings (including multiple brain areas within the frontoparietal network[55–61]) will almost certainly be necessary to fully characterize the processes leading to movement decisions.

## Methods

**Subjects**. We trained two male rhesus macaque monkeys (*Macaca mulatta*) to make reaches using a planar robotic manipulandum for water or juice reward. All procedures were approved by the Northwestern University Institutional Animal Care and Use Committee (Protocol number #IS00000367) and were consistent with the Guide for the Care and Use of Laboratory Animals.

**Behavioral task**. The monkeys sat in a chair facing a vertical monitor and used a planar robotic manipulandum to control an on-screen cursor. Each 2-Target trial (40%) began once the monkey held the cursor within a central target (1.5 cm radius) for 500 ms, after which two outer targets appeared (Target On; 750–1000 ms), always 180° apart. The target locations were restricted to eight different locations, equally spaced at a radius of 7 cm. After the Target On period, the outer targets briefly disappeared (Target Blank; 250–500 ms). After the Target Blank period, one of the targets reappeared (Cue; 250–500 ms), providing the monkey with complete information of the correct target. The missing—i.e., incorrect—target then reappeared, returning to the initial 2-Target presentation. The center

target was extinguished, and a tone signaled the monkey to move (Go; <5 s). The trial ended when the cursor reached one of the outer targets. If the target was correct (as indicated by the target presented during the Cue period), the monkey received a success tone coincident with delivery of a liquid reward. If incorrect, the monkey received a failure tone and the screen displayed the location of the correct target.

1-Target trials (40%) followed the same structure as above, except only the correct target was ever shown. Thus, the monkey had complete information about the correct target from the beginning of the trial. On 20% of all trials, we omitted the Cue period altogether. On these Free-Choice trials (10%), the monkey was forced to choose one of the two targets without any information as to which was correct. We maintained the same reward structure during these trials, so there was always a 50% chance of reward. 1-Target trials without a Cue period were not used in any analyses.

**Neural recordings and preprocessing**. Both monkeys were implanted with two chronic, 96-electrode arrays (Blackrock Microsystems, Salt Lake City, UT) positioned over the arm area of primary motor cortex (M1; 1.5 mm electrode length) and straddling the rostral/caudal division of PMd (1.0 mm electrode length). We discriminated single neurons offline by isolating clusters within a principal component space calculated from the waveform shapes of putative neurons (Plexon Inc., Dallas, TX). For monkey C this yielded unit counts of 143, 143, and 108. For monkey M: 154, 132, and 114.

We used the spiking events from each recorded unit to calculate a continuous estimate of firing rate by first convolving with a half-Gaussian kernel (s.d. 150 ms) and then downsampling to 40 Hz. We chose a half-Gaussian kernel to ensure a purely causal relationship between spiking events and the estimated firing rate. We then applied a square root transform to all firing rate estimates.

**Trial-averaged activity traces**. The trial-averaged activity traces from Fig. 2a, b included all correct trials for both monkeys. We first identified the preferred direction for each neuron by fitting a simple cosine model to 1-Target planning activity (500 ms post-Target On to the end of Target Blank). Then for each trial, we separated neurons into pro-PD (preferred direction within 22.5° of reach direction), anti-PD, and orthogonal-PD. We then performed a 1000-fold bootstrap on the difference between pro-PD and orthogonal-PD (and between anti-PD and orthogonal-PD) to obtain 95% confidence bounds.

**Dual representation vs. single representation: single-neuron control analysis**. We based the control analysis performed in Fig. 2c, d, e on an analysis described by Cisek and Kalaska[11]. We first calculated the preferred direction for each neuron using planning activity on 1-Target trials and a simple cosine model. To allow for the possibility of dual representation, we then fit a cosine model with doubled frequency to 2-Target activity. This resulted in bimodal curves—corresponding to equal responses for each target pair—with the two peaks defining the preferred axis of each neuron. For very model fit, we determined significance through a bootstrap procedure. We first fit the appropriate cosine or doubled-cosine model and recorded the amplitude term (i.e., modulation depth). Then, for each of 1000 bootstrap samples, we scrambled the firing rates with respect to reach direction and repeated the fit. Only neurons for which the amplitude exceeded 99% of the scrambled fit amplitudes were considered significantly tuned and included in the analysis. Additionally, we discarded neurons for which the preferred direction (single cosine) and preferred axis (doubled cosine) differed by more than 30°.

For each neuron deemed significantly tuned, we compiled all planning activity (500 ms post-Target On to the end of Target Blank) on pro-PD, anti-PD, and orthogonal-PD trials. We then calculated the average number of time bins for which pro-PD and anti-PD activity exceeded the median orthogonal-PD activity.

**Dual representation vs. single representation: guessing simulation**. The simulations of the unbiased guessing and biased guessing results in Fig. 2d, e involved only relabeling reach directions on 1-Target trials. For unbiased guessing, we relabeled half of the trials for each reach direction as corresponding to the opposite direction. This resulting dataset simulated the case in which the monkey might initially create a plan to one of the two targets on each trial, chosen randomly. To simulate the biased guessing paradigm, we performed a similar relabeling procedure as before, but first discarded 1-Target trials in four of the eight directions (counterclockwise, starting with the left target). We then randomly relabeled half of the remaining trials as corresponding to the opposite direction. The resulting dataset simulated the case in which the monkey might begin each trial by planning to a specific target of the pair (either the right, upper-right, upper, or upper-left target).

**Dimensionality reduction and reach plan decoding**. On each session, we grouped all 1-Target firing rates into the matrix $M \in \mathbb{R}^{N \times T}$, where $N$ was the number of neurons and $T$ was the total number of time points after concatenating all 1-Target trials. We then performed PCA on the matrix $M$ and projected all firing rates (from 1-Target, 2-Target, and Free-Choice) onto the top 10 principal axes, defining a 10-dimensional neural state.

We then calculated, at every time point, a proximity of the neural state $S$ to the reach planning/execution states for each direction. For each 1-Target reach direction $i$, we assembled all neural states (from 500 ms after Target On until trial completion, all trials) into the set $\{C_i\}$. For every time point, we calculated the proximity of $S$ to the neural states corresponding to activity for a given reach direction $i$ as follows:

$$\text{Proximity}(S, \{C_i\}) = \frac{P(D_M(S, \{C_i\})|i)}{\sum_{j=1}^{8} P(D_M(S, \{C_i\})|j)}$$

where $D_M(S, \{C_i\})$ represents the Mahalanobis distance between the point $S$ and cluster $\{C_i\}$. The metric uses probabilities based on empirical distributions of Mahalanobis distances within (numerator) and across (denominator, $j \neq i$) reach directions, which acts as a normalization. While the range of the raw Mahalanobis distances calculated for each reach direction could vary considerably, the proximity metric is limited to the range [0, 1], allowing for a fairer comparison of proximities to different target clusters. Although Proximity as defined is technically a probability, it is conditioned only on distances to a single-reach planning/execution state. Thus, the Proximities calculated for all directions do not sum to one and cannot not be interpreted as probabilities in a meaningful way.

We chose to use the difference in Proximities ($\Delta$Proximity) rather than the output of a full classifier to track time-varying reach plans. This decision ensured that decoded plan "strength" depended on absolute distance of the instantaneous neural state $S$ to the assembled reach-related planning/execution states, and not relative distance. For example, consider a neural state very far away from all training sets $\{C\}$. Unless that state was almost perfectly equidistant between all sets, its relative (and entirely coincidental) nearness to one could cause a probabilistic classifier to return a highly confident classification. However, since the state was not actually close to any cluster within the training set, the $\Delta$Proximity metric would return a value very close to zero (see Supplementary Figure 3). Thus, the $\Delta$Proximity metric is more conservative than the output of a classifier and returns large magnitudes only when a neural state can be confidently assigned to a specific reach state within the training set. For 2-Target trials, we calculated $\Delta$Proximity with respect to the two presented targets. On 1-Target trials, we calculated with respect to the presented target and the opposite target.

**Neural state simulations**. To simulate activity according to the dual-target model, we began by fitting a cosine with 1-Target reach directions $\theta$ and average Target Blank firing rates ($F$) for each neuron:

$$F(\theta) = a + b \cdot \cos(\theta - \varphi)$$

We then constructed an artificial bimodal tuning curve $F_{bi}(\theta)$ for each neuron using the previously obtained values of $a$, $b$, and $\varphi$:

$$F_{bi}(\theta) = 0.7 \cdot [a + b \cdot \cos(2(\theta - \varphi))]$$

We used these bimodal tuning curves to simulate the expected activity of every neuron on a given trial. Note that we included a 0.7 scaling factor to match the average difference in firing rates between 1- and 2-Target trials as reported by Cisek and Kalaska[11]. However, changing this factor had no qualitative effect on the neural state analyses.

To simulate the averaged-plan model, we first calculated nonparametric 1-Target tuning curves $F_{nonpar}$ by averaging the Target Blank activity for each reach direction. We then simulated responses for each direction $\theta_i$:

$$F_{average}(\theta_i) = \frac{1}{2}\left(F_{nonpar}(\theta_i) + F_{nonpar}(\theta_i + \pi)\right)$$

We simulated stay-or-switch responses simply using $F_{nonpar}$. For all simulations, we omitted additional noise to avoid introducing additional factors that might affect the neural state analyses. As a result, for each model we could only simulate one population response for each target pair (or for each target in the case of the stay-or-switch model).

**Dual representation index**. Here we aimed to construct a single-valued metric on the Proximity space (as seen in Figs. 3–5) to distinguish between dual representations and single representations. We defined the DRI on two Proximities Prox $(S,a)$ and Prox$(S,b)$ as:

$$\text{DRI} = \sqrt{\frac{[\text{Prox}(S,a)]^2 + [\text{Prox}(S,b)]^2}{2}} \cdot \left(\frac{\text{Prox}(S,a) + \text{Prox}(S,b)}{\max(\text{Prox}(S,a), \text{Prox}(S,b))} - 1\right)$$

The formula for DRI involves the multiplication of two terms. The first term returns higher values for points farther from the origin, and the second for points close to the diagonal. The final DRI metric is approximately equal to min(Prox(S, a), Prox(S,b)) but the second term skews it to higher values for points close to the

diagonal. This property makes it better able to differentiate between dual-target or averaged-plan representations (points close to unity; Fig. 4c, f) and single-target representation states (points close to the axes; Fig. 4i).

**Reaction time correlation**. We defined reaction time as the time elapsed from the Go cue to the time at which the hand speed exceeded 5 cm s$^{-1}$. The average peak speed was 23.9 cm s$^{-1}$, with 95% falling between 12.7 and 43.9 cm s$^{-1}$.

We used a linear mixed effects model to test for a general correlation between reaction time and $\Delta$Proximity across all reach directions and sessions. We used the fitlme function in Matlab (Mathworks) with reaction time as the response variable, $\Delta$Proximity as the predictor variable, and reach direction (separated by session) as the grouping variable, with uncorrelated random effects for intercept and $\Delta$Proximity.

**Code availability**. Custom Matlab code used in the manuscript can be provided upon reasonable request.

## Data availability
Relevant data will be made available for further analysis upon request.

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

## Acknowledgements
The authors gratefully acknowledge the helpful discussions with Aaron Batista and with Paul Cisek as we sought to compare our results to earlier published results.

## Author contributions
B.M.D. contributed to task design, data acquisition, analysis, and interpretation, and drafting and revising the article. K.P.K. and L.E.M. contributed to interpretation of results and drafting and revising the article.

## Additional information

**Competing interests:** The authors declare no competing interests.

