## [Peer Review File · Nature Communications]

Reviewers' comments:

Reviewer #1 (Remarks to the Author):

General comments:

This study by Dekleva et al. tries to answer the question if multiple potential reach plans are simultaneously planned in the cortex. The paper is generally well written and uses a new approach to tackle the question of goal representation in the brain. However, since the authors make a pretty strong claim about the cortical mechanism of goal planning which is controversial to several papers (which they do cite and discuss) their interpretation and methods have to be scrutinized carefully. While reading the manuscript several questions came up about the methods, the results and their interpretation which I think need clarification before I would feel comfortable to recommend it for publication.

Major issues and questions:

1. Even though the authors did only record from a relatively small part of PMd they make a very strong claim in the abstract and conclusion of the manuscript: "Our results suggest that cortex plans only one option at a time...". One of the studies they cited, Cisek & Kalaska 2005, reported a caudal-rostral gradient of target specificity. More rostral areas would contain more cells that are active for potential reach goals. Could it be that the array placement was too close to M1 to yield similar results as reported previously? I find it a bit hasty to conclude the cortex would only ever plan one option at a time considering that there have been studies in different parts of premotor cortex and parietal including a whole theory about how more vague plans are transformed to more concrete plans the closer to motor one gets.
2. I think the analysis of the neural state is a very interesting and important point of the paper. Still for comparison sake it would have been useful to also provide a more in-depth analysis of single cell selectivity. In the supplementary material one single cell analysis is performed but it would have been more informative to show single cell behavior in more detail.
3. Interestingly even though they recorded in PMd and M1 results from M1 seem to be completely omitted in the manuscript. Why was the same analysis not applied to the M1 data that they must have recorded? This would have helped to compare their results better to previous studies that they cited.
4. One of the main points of the paper is shown in figure 4a. Here the authors argue that 88% of all trajectories are biased for a movement in one direction during the blank period. Looking at the start of the cue period it looks as if this ratio must have changed near the end of the blank period. Since the blank period was variable in length it would have been interesting to see how the trajectories would look like if they were aligned to the cue and then tracing them backwards to understand what happened at the end of the blank period. Even in those traces they show in the blank period it looks as if a shift happens towards 0 likelihood. What is the explanation for this? Also, why not show a plot like 4a for all directional pairs and all sessions?
5. The authors mention that in other experiments it has been tried to unbiased the monkeys. Has there been any attempt to try this and see if the neural state trajectories would change then?
6. This paper heavily depends on comparisons to Cisek & Kalaska 2005 (and others). If the main point was to reevaluate those results why hasn't the task paradigm be exactly replicated? One of the problems I see here is for example that the important blank period is kept rather short in this experiment (250-500 ms vs. more than 1000 ms in other cited papers).
7. Since the strategies the monkeys use and the biases for specific directions depends a lot on the training history it would be interesting to know how the training was done and how many training sessions had been done before the analysis phase.
8. Since the claim is made that the cortex plans only one reach option at a time how are those trajectories interpreted which are close to 0 likelihood? There are clearly some cases when the bias was only very weak or not present at all.

9. What was the total number of recording sessions and trials per session? It sounds as if there were only three sessions which is rather low especially considering that there might be training effects which could change the strategies the monkeys were using and the neuronal representation.

Minor issues:

1. Page 2 19-21: Some of those studies did not record from motor cortex. In fact most of the cited studies are recording from premotor or parietal.
2. Figure 2 figure caption (b): ...the neurons from a during... Sentence?
3. Figure 4 figure caption (a): Which ones are cued for right / left?
4. Page 19 252-253: Why three unit counts? Is that because there were only three recording sessions? If so this sounds like rather few.

Reviewer #2 (Remarks to the Author):

In this paper, Dekleva and colleagues examine whether multiple actions can be prepared at the same time. Previous studies have found that neurons encode both potential upcoming movements when multiple targets are cued (e.g. Cisek and Kalaska, 2005). The authors accurately point out that these findings could be caused by two possible models: either neurons actually signal both reach targets simultaneously, or neurons encode one or the other on individual trials (which could falsely yield a signal for "both" when neural signals are averaged across trials).

To address this question, the authors recorded neural activity using multi-channel arrays while monkeys made reaches cued by either two targets or one target. They then performed single-trial analysis in order to assess whether the monkeys were preparing movements to one target or the other, over time.

The question that this paper sets out to address is quite interesting. Figuring out whether motor plans can be performed simultaneously or whether they change plans on a trial by trial basis or over time has important implications for how the motor cortex functions. Furthermore, this question can indeed be best addressed by careful analysis of single trials, as the authors propose.

However, I have serious reservations about the analytical methodology used to address this question. As such, I have little confidence in the results as presently stated. I do think that the experiment itself was well-designed, and the authors' data ought to be able to speak to this important question. However, major revisions would be required before I could have any serious confidence in their results.

Major comment 1: My primary concern is associated with Figure 2. Given that the paper seeks to demonstrate that motor cortex does not simultaneously prepare reaches to two targets, this figure becomes of paramount importance: it is the only figure that directly assesses this question. The overall message of the figure is that encoding two reach directions simultaneously will sufficiently break the correlation structure of the units as to result in completely different subspaces encoding movements in one-target versus two-target tasks (as implied by the statement on line 67-68 "Such nonstationarity would invalidate the dimensionality reduction-based approach"). I find this claim hard to believe.

The simplified model reported in Figure 2a-c does indeed show the effect that the authors describe. However, if modified even slightly, the core result does not hold. For example: if the targets are not 180 degrees apart, then the "two target" covariance structure begins to contain a high degree of overlap with the "one target" covariance structure. For a second example: if the units are allowed a

more complicated FR tuning (rather than being perfectly cosine tuned), then there will also be more overlap.

It is also unclear why the authors chose to model a neuron's "two target" response as the maximum amount that it would fire for either target. Why not fire the average (or the sum) of their responses to the two targets? I can only assume that this was because the average in a perfectly cosine tuned model would leave a completely flat curve for the 180 degree separated case. The authors should either show that their predictions are robust to multiple models of how neurons might simultaneously encode two targets, or offer strong justification why this is the correct model.

One way that the authors may be able to demonstrate the presence (or absence) of new dimensions for encoding two targets simultaneously would be to run their analysis on trial-averaged data for two-target trials. By trial averaging, they would "build in" the simultaneity (as they point out in their introduction). They would then have data to say if there really were simultaneous encoding, how overlapping would the spaces be, given the actual firing properties of the units? If the spaces truly are highly different between the two cases, then they could examine single trials to see if the effect holds. I would also suggest in general that they spend more time and figure space talking about the simulations they ran in 2e, as the simulations which begin with the real data are likely to have more relevance to the system than the toy model.

Major comment 2: I am somewhat concerned about the method used to decode which target is being prepared at any given time. The authors build a training set from all single-target trials, from 500 ms post-target until trial completion, grouping all reaches to each target together. For each two-target trial, they then calculate the Mahalanobis distance to each of these groups at each time. I have two major concerns about this analysis:

My first concern is that this decoding analysis as presented may not do a good job at telling the difference between a "simultaneous" versus "alternating" encoding scheme. For example, consider a model in which there is a sub-population of neurons which always fires if a left target is present, and a separate sub-population which always fires if a right target is present. Looking at the neural distance in the "two target" case, we would find the distance to be high compared to the "one-target" rightward trials (because there is a lot of activity in the "left" units, which would drive distances up). Distance would also be high compared to the leftward trials. This would likely result in a low probability for both targets. This issue is not unique to this toy "subpopulation" model I describe: it would also be an issue if different neural dimensions corresponded to different reach directions, for example. The authors state in their methods that they calculate probabilities independently for each direction to allow for the possibility of dual or intermediate reach plans during two-target trials. However, nowhere do they compare, for example, the absolute probability calculated for two-target versus for one-target trials. Is the confidence significantly lower when multiple targets are present? Or is it the same? Just looking at the relative probabilities for each target is unlikely to be conclusive: the absolute probability is also relevant.

My second concern is that, in effect, this analysis is computing the neural distance between single trials and a single, non-time-varying, average neural state for each target (Mahalanobis distance calculates the distance to the center of the distribution of single-trial states, across all trial times). This may not be a sophisticated enough model to truly get at the question. In particular, neurons' tuning can change dramatically between preparation and movement epochs (Churchland et al, Neuron 2010), so by averaging across these epochs and attempting to find a single "average state," you may be missing some important changes of activity that occur in the transition from preparation to movement. This could cause your time-varying estimations of probability to be inaccurate (and

potentially drive spurious fluctuations between which reach is most probable). At the very least, I would recommend restricting the analysis to plan-related activity and excluding activity around and after movement onset. Alternately, the authors could examine activity in a manner that takes varying neural activity over time into account.

In general, I would suggest that the authors consider the different ways that neural activity should behave over the course of a single trial in the "simultaneous" and "non-simultaneous" encoding cases. Then, use a single-trial analysis tool that can distinguish between those cases, and report ways in which the neural data resembles one or the other predictions. Without clear expectations of what the single-trial decodes should look like in the case of two targets encoded simultaneously versus non-simultaneously, it is hard to use the data from figure 3 onward to draw strong conclusions.

Minor comments:

Figure 3b: Please include a scale bar

Figure 4c: This figure is hard to interpret. The figure text reads "Each point reflects the neural preference for the given axis on a particular session." My interpretation would then be that (for example), on the horizontal axis, a dot to the right indicates a preference for rightward plans, where a dot to the left indicates a preference for leftward plans, with the radius indicating the magnitude of preference. Why, then, is the 50% mark not at the center of the circle?

Discussion: Line 187-188 reads: "The high degree to which the initial reach plan predicted the free-choice behavior suggests that early responses in PMd strongly influence the eventual movement decision." This sentence is overly strong. Activity in PMd could derive from a decision being made elsewhere, reflecting (but not driving) a bias toward the initial choice. Please re-phrase this sentence to take this correlation-vs-causation issue into account.

Methods: Line 295-296 reads "We then used the Mahalanobis distance between the neural state and the training set..." I assume that the training set is the set of all single-target reaches described earlier in the paragraph, but this is not explicitly stated.

In general, the figure text contains many minor errors. Please re-read the figure labels closely and ensure that they are accurate. Below are the issues I found:

Figure 4a: Figure text reads "Blue traces indicate those for which the monkey was cued to the right. Red traces indicate a cue to the right." I assume one of these colors should correspond to left.

Figure 5: In the figure text, (d) should instead be (c).

Figure 6c: Currently reads "As in a for all reaches to the upper left target..." I assume this should be for the upper right target, based on the figure inset highlighting upper right reaches.

Reviewer #3 (Remarks to the Author):

In this article, Dekleva and colleagues examine preparation of reaching movements in conditions of multiple potential targets. They argue that only a single reach plan is ever prepared prior to movement - contrary to prominent and prevailing theories suggesting that the motor system prepares multiple movement plans in parallel.

The manuscript is clearly written and overall I believe the results do support the claim of a single movement plan. However I do have a number of significant concerns about weak organization and

logic of the paper.

Major Concerns:

1) The main evidence supporting the claim of a "single reach plan" is the comparison of the covariance in the 1- versus 2-target conditions. Oddly, this analysis is not framed as being the critical test of the single-plan hypothesis, but rather is framed as a validation of being allowed to do dimensionality reduction. The rest of the manuscript after this key result then proceeds to explore the phenomenology of the task (errors, reaction times), with the assumption that the monkey is maintaining only a single movement plan. While these additional observations are interesting, it isn't clear that they provide any additional evidence in favor of a single reach plan.

On a related note, I am not convinced that the 'fixed correlation' requirement for dimensionality reduction is really applicable. line 59 - "For this dimensionality reduction approach to be valid, the correlations between neurons must be fixed". I don't follow why this is the case. In general, it doesn't seem to me that individual neurons need to have a fixed pairwise correlation for dimensionality reduction to be valid. Typically, pairs of neurons would have a fixed correlation during repeated movement to the same target, but would be differently correlated for different reach directions. Applying the authors' logic, one should then be prohibited from doing dimensionality reduction for tasks with multiple targets, which is of course absurd. In the example provided by the authors, it seems perfectly valid to employ a dimensionality reduction in which $d1 = (\text{neuron1} + \text{neuron2})$ and $d2 = \text{neuron3}$. I fail to see what the problem would be with this.

2) Regarding the hypothesis itself - The authors neglect to mention that several authors have argued in the past that only a single movement plan exists. For instance, Haith et al. (2015) and Wong and Haith (2017) argued on the basis of behavioral evidence that the motor system only ever maintains a single movement plan. Also, Churchland et al. (Neuron, 2010; Nature 2012) proposed that the movement being prepared is determined solely by the initial state of the motor cortex (primarily PMd) at the time of movement initiation. Implicit in this theory is that it is not possible for the motor cortex to simultaneously represent two or more distinct movement plans.

3) The decoding algorithm used to estimate the likelihood is unprincipled and inadequately described. $P(i|D_M(S, \{C_i\}))$, in the equation on line 298, looks like it is supposed to represent a probability (not a likelihood, as claimed in the text - technically the likelihood of target i would simply be $P(S|i)$). However, the term computed is not really a probability - the "probabilities" do not add to one. Properly turning this into a probability would involve applying Bayes' rule (taking $P(i)$ into account). The LHS of line 298 doesn't really correspond to any readily interpretable quantity and, in the end, it feels like an ad hoc metric that seemed to work, rather than the product of a principled approach to inferring the reach plan from the neural data.

It is not clear what is meant by $P(D_M(S, \{C_i\})|i)$. How exactly is a probability computed based on the distance? I presume the idea is that $P(S|i)$ has a multivariate normal distribution, in which case $P(S|i)$ depends on the Mahalanobis distance (it would be proportional to $\exp(-D_M(S, \{C_i\})^2)$). Note that the use of the Mahalanobis distance is actually redundant with conditioning on i . Computed properly, $P(S|i)$ should already involve computation of the Mahalanobis distance.

It is not at all clear what the denominator, $P(D_M(S, \{C_i\})|j)$ corresponds to. What does it even mean to condition on j in this case? Does this mean the probability of observing that Mahalanobis distance under the generative model for target j ? If so then why the Mahalanobis distance for i ? What is the rationale for doing this normalization at all? Why not just use max likelihood estimation?

It's not clear why it's necessary to perform dimensionality reduction down to 10 dimensions before doing this analysis. The Mahalanobis distance would naturally ignore (or give a very low weight to) the dimensions with low variance that are discarded during the PCA.

4) An important contribution of the paper is to explain how previous studies could have come to a spurious conclusion of parallel planning. The supplemental materials therefore constitute a critical component of the paper (in my view, much more critical than the bulk of what appears in the main paper). However, these analyses are not explained in sufficient detail. It is not possible to follow the details without referring to the Cisek and Kalaska paper. In particular, the key measure, plotted on the x-axis of Fig S1c,d: "% of PD+OD activity that is greater than the median of orthogonal activity" is not explained at all. Even "OD" is not defined anywhere. Given the importance of this argument to the overall claim, I feel substantially more care is warranted to make this analysis and argument clearer and self-contained (i.e. not needing to refer to Cisek/Kalaska to follow it completely).

Regarding the outcome of the analysis, if I have read Figure S2d correctly, it seems to suggest that the existence of a target preference can lead to a conclusion of parallel planning even after controlling for target preferences in the 2-target condition (the distribution for the 60% target preference is shifted above 50%). How can this be reconciled with Figure S1d, which shows that, for the actual data, no such shift happens? Does this imply that the stay-or-switch model used in the simulation isn't an accurate model of what really happened in the task?

Additional Comments:

Was there any relationship between plan "strength" and movement speed? If so, this might suggest that both RT and tuning strength might both be consequences of the monkey's motivational state, rather than the RT being a consequence of the preparatory state.

What is the rationale for taking the square root of firing rate?

hyphenation errors throughout:

line 31 ("a multiple-potential-target reaching task")

line 121 ("single reach plans")

line 166 ("two-target tasks")

line 230 ("target-on period")

line 232 ("target-off period")

line 240 ("One-target trials")

line 261 ("the desired-reach-direction-tuning-related covariance") - This one is a bit much even when correctly hyphenated. might be worth rephrasing.

line 294 ("post-target-appearance")

line 234 - "...the correct TARGET".

Response to Reviewers

We wish to thank the reviewers for their many insightful comments. We have attempted to address all that were feasible. Unfortunately, as neither monkey is still available, we are unable to add any more data to the study.

Reviewer #1

Major issues and questions

1. Even though the authors did only record from a relatively small part of PMd they make a very strong claim in the abstract and conclusion of the manuscript: “Our results suggest that cortex plans only one option at a time...”. One of the studies they cited, Cisek & Kalaska 2005, reported a caudal-rostral gradient of target specificity. More rostral areas would contain more cells that are active for potential reach goals. Could it be that the array placement was too close to M1 to yield similar results as reported previously? I find it a bit hasty to conclude the cortex would only ever plan one option at a time considering that there have been studies in different parts of premotor cortex and parietal including a whole theory about how more vague plans are transformed to more concrete plans the closer to motor one gets.

We acknowledge that our findings cannot (and should not) be extrapolated to all of cortex, and we have reflected that limitation in the text. However, based on the array placements (Fig 1) and prominence of planning-related activity in the averaged responses (Fig 2) we are reasonably confident that the recordings were made from dorsal premotor cortex. We did not observe any difference in response characteristics between the rostral and caudal halves of the array, likely due to the small region of cortex it covered.

2. I think the analysis of the neural state is a very interesting and important point of the paper. Still for comparison sake it would have been useful to also provide a more in-depth analysis of single cell selectivity. In the supplementary material one single cell analysis is performed but it would have been more informative to show single cell behavior in more detail.

We have included trial-averaged results of single-neuron activity in a new figure (Fig 2). However, due to space constraints, we have chosen not to show single-neuron rasters, tuning curves, etc. As discussed in the text, the trial-averaging required to describe single-neuron responses may lead to misleading results. We believe that the state-space analyses (which have been significantly clarified) are most appropriate for addressing the main question of single- or dual-representation.

3. Interestingly even though they recorded in PMd and M1 results from M1 seem to be completely omitted in the manuscript. Why was the same analysis not applied to the M1 data that they must have recorded? This would have helped to compare their results better to previous studies that they cited.

We have now included M1 results in the supplementary section (Fig S1). M1 exhibited less planning-related activity, but the state-space analysis results were almost identical to those from PMd.

4. One of the main points of the paper is shown in figure 4a. Here the authors argue that 88% of all trajectories are biased for a movement in one direction during the blank period. Looking at the start of the cue period it looks as if this ratio must have changed near the end of the blank period. Since the blank period was variable in length it would have been interesting to see how the trajectories would look like if they were aligned to the cue and then tracing them backwards to understand what happened at the end of the blank period. Even in those traces they show in the blank period it looks as if a shift happens towards 0 likelihood. What is the explanation for this? Also, why not show a plot like 4a for all directional pairs and all sessions?

We have realigned the traces to the Cue to make trends more apparent. The tendency of the decoded plan strength to drift towards 0 after target disappearance is not consistent across sessions. It is therefore difficult to

determine whether the apparent “weakening” is a noteworthy effect. To address the point that we showed only one target pair, we now include plots that incorporate data from all trials and all sessions.

5. The authors mention that in other experiments it has been tried to unbiased the monkeys. Has there been any attempt to try this and see if the neural state trajectories would change then?

Unfortunately, the monkeys in this study are no longer available. However, we included as a discussion point the effect “unbiasing” might have on planning responses. We also discuss the implications of training, and whether the subsequent development of a neural effect would reflect natural sensorimotor control or something like specialized skill learning.

6. This paper heavily depends on comparisons to Cisek & Kalaska 2005 (and others). If the main point was to reevaluate the those results why hasn't the task paradigm be exactly replicated? One of the problems I see here is for example that the important blank period is kept rather short in this experiment (250-500 ms vs. more than 1000 ms in other cited papers).

The simplicity in task design allowed us to evaluate “natural” planning behavior on a two-target task while limiting potential effects of overtraining. We intended to later transition to the exact version used by Cisek and Kalaska (2005), but were unsuccessful due to monkey behavioral issues and then array failure. The similarity in trial-averaged activity between our results and those from Cisek and Kalaska (2005) suggest that our task did induce similar neural responses. It seems likely that the major results reported in the literature (e.g., dual representation) would be consistent across task variants.

7. Since the strategies the monkeys use and the biases for specific directions depends a lot on the training history it would be interesting to know how the training was done and how many training sessions had been done before the analysis phase.

The monkeys were already trained on a simple 1-Target center-out task. For the present study, they were shown the 2-Target task as described until they could perform at an above-chance level. This took about 3 sessions for Monkey C and 2 sessions for monkey M. We did not find the directional preferences to be consistent or follow a clear pattern. We now also comment on the potential effect of training on the representation of multiple reach targets in the discussion section.

8. Since the claim is made that the cortex plans only one reach option at a time how are those trajectories interpreted which are close to 0 likelihood? There are clearly some cases when the bias was only very weak or not present at all.

We have included new figures (Figs 3,5) that should help address this question. In brief, decoded magnitudes near 0 should be interpreted as “no reach plan” rather than “two opposing and equally strong reach plans”.

9. What was the total number of recording sessions and trials per session? It sounds as if there were only three sessions which is rather low especially considering that there might be training effects which could change the strategies the monkeys where using and the neuronal representation.

Each monkey completed 3 full sessions (after training) of the task as described. Sessions averaged 400 trials (min: 227, max: 560). While the session count was a bit low, the amount and quality of recorded neurons, and the consistency across sessions and monkey appear to be sufficient to answer the questions posed. We include the question of trained neuronal representations as a discussion point.

Reviewer #2

Major issues and questions

1. My primary concern is associated with Figure 2. Given that the paper seeks to demonstrate that motor cortex does not simultaneously prepare reaches to two targets, this figure becomes of paramount importance: it is the only figure that directly assesses this question. The overall message of the figure is that encoding two reach directions simultaneously will sufficiently break the correlation structure of the units as to result in completely different subspaces encoding movements in one-target versus two-target tasks (as implied by the statement on line 67-68 “Such nonstationarity would invalidate the dimensionality reduction-based approach”). I find this claim hard to believe.

We acknowledge that the analysis did not adequately discriminate between single-target and dual-target representations. We have removed the previous figure and added new analysis that more clearly distinguishes between the planning models.

The simplified model reported in Figure 2a-c does indeed show the effect that the authors describe. However, if modified even slightly, the core result does not hold. For example: if the targets are not 180 degrees apart, then the “two target” covariance structure begins to contain a high degree of overlap with the “one target” covariance structure. For a second example: if the units are allowed a more complicated FR tuning (rather than being perfectly cosine tuned), then there will also be more overlap.

It is also unclear why the authors chose to model a neuron’s “two target” response as the maximum amount that it would fire for either target. Why not fire the average (or the sum) of their responses to the two targets? I can only assume that this was because the average in a perfectly cosine tuned model would leave a completely flat curve for the 180 degree separated case. The authors should either show that their predictions are robust to multiple models of how neurons might simultaneously encode two targets, or offer strong justification why this is the correct model.

For simulations of dual representation, we now include two different models. One is based on 1-Target and 2-Target tuning differences reported by Cisek and Kalaska (2005), and the other is based on the “averaged target” model as described. Both models lead to essentially the same conclusions.

One way that the authors may be able to demonstrate the presence (or absence) of new dimensions for encoding two targets simultaneously would be to run their analysis on trial-averaged data for two-target trials. By trial averaging, they would “build in” the simultaneity (as they point out in their introduction). They would then have data to say if there really were simultaneous encoding, how overlapping would the spaces be, given the actual firing properties of the units? If the spaces truly are highly different between the two cases, then they could examine single trials to see if the effect holds. I would also suggest in general that they spend more time and figure space talking about the simulations they ran in 2e, as the simulations which begin with the real data are likely to have more relevance to the system than the toy model.

We appreciate these suggestions and have incorporated them into the new figures, specifically the “averaged-target” simulations in Figure 4d-f.

2. I am somewhat concerned about the method used to decode which target is being prepared at any given time. The authors build a training set from all single-target trials, from 500 ms post-target until trial completion, grouping all reaches to each target together. For each two-target trial, they then calculate the Mahalanobis distance to each of these groups at each time. I have two major concerns about this analysis:

My first concern is that this decoding analysis as presented may not do a good job at telling the difference between a “simultaneous” versus “alternating” encoding scheme. For example, consider a model in which there is a sub-population of neurons which always fires if a left target is present, and a separate sub-population which always fires if a right target is present. Looking at the neural distance in the “two target” case, we would find the

distance to be high compared to the “one-target” rightward trials (because there is a lot of activity in the “left” units, which would drive distances up). Distance would also be high compared to the leftward trials. This would likely result in a low probability for both targets. This issue is not unique to this toy “subpopulation” model I describe: it would also be an issue if different neural dimensions corresponded to different reach directions, for example. The authors state in their methods that they calculate probabilities independently for each direction to allow for the possibility of dual or intermediate reach plans during two-target trials. However, nowhere do they compare, for example, the absolute probability calculated for two-target versus for one-target trials. Is the confidence significantly lower when multiple targets are present? Or is it the same? Just looking at the relative probabilities for each target is unlikely to be conclusive: the absolute probability is also relevant.

We have added scatterplots (Figs 3-5) to show the absolute values (now *Proximity*, previously *Likelihood*) calculated for the two relevant reach directions on each trial. This has removed much of the ambiguity caused previously by reporting only the difference. We have also included simulations of single- and dual-target representations to demonstrate the expected Proximity values for various models of dual- and single-target planning activity (Fig 4). We believe these additions greatly clarify our results and improve interpretability.

My second concern is that, in effect, this analysis is computing the neural distance between single trials and a single, non-time-varying, average neural state for each target (Mahalanobis distance calculates the distance to the center of the distribution of single-trial states, across all trial times). This may not be a sophisticated enough model to truly get at the question. In particular, neurons’ tuning can change dramatically between preparation and movement epochs (Churchland et al, Neuron 2010), so by averaging across these epochs and attempting to find a single “average state,” you may be missing some important changes of activity that occur in the transition from preparation to movement. This could cause your time-varying estimations of probability to be inaccurate (and potentially drive spurious fluctuations between which reach is most probable). At the very least, I would recommend restricting the analysis to plan-related activity and excluding activity around and after movement onset. Alternately, the authors could examine activity in a manner that takes varying neural activity over time into account.

While these are reasonable concerns, we found that incorporating the entire trial in our computation did not decrease our ability to decode from individual epochs. Furthermore, restricting the analysis to use only data from the same epoch did not improve decoding from that epoch (judged by decoding accuracy on 1-Target trials) and made it difficult to view transitions between epochs. While single-neuron tuning does change throughout a trial, these issues did not appear to affect calculations in the reduced (but still high-dimensional [10D]) space. A more sophisticated computation incorporating dynamics would likely improve decoding. However, multiple pieces of evidence (decoding accuracy on 1-Target and free-choice trials, correlation with behavioral biases, and agreement between brain areas) suggest that the decoded movement plans were not artifacts of inappropriate analysis.

In general, I would suggest that the authors consider the different ways that neural activity should behave over the course of a single trial in the “simultaneous” and “non-simultaneous” encoding cases. Then, use a single-trial analysis tool that can distinguish between those cases, and report ways in which the neural data resembles one or the other predictions. Without clear expectations of what the single-trial decodes should look like in the case of two targets encoded simultaneously versus non-simultaneously, it is hard to use the data from figure 3 onward to draw strong conclusions.

We thank the reviewer for this suggestion and have included analyses and figures accordingly. The simulations in Figure 4 should establish clear expectations for single- and dual-target representations. Also, the Dual Representation Index metric reported in Figure 6 provides a straightforward comparison between observed data and simulated models of dual representation.

Minor comments:

- 1. Page 2 19-21: Some of those studies did not record from motor cortex. In fact most of the cited studies are recording from premotor or parietal.**
- 2. Figure 2 figure caption (b): ...the neurons from a during... Sentence?**

3. Figure 4 figure caption (a): Which ones are cued for right / left?

4. Page 19 252-253: Why three unit counts? Is that because there were only three recoding sessions? If so this sounds like rather few.

We have made significant changes to the text that have addressed these issues.

Figure 3b: Please include a scale bar

Figure removed

Figure 4c: This figure is hard to interpret. The figure text reads “Each point reflects the neural preference for the given axis on a particular session.” My interpretation would then be that (for example), on the horizontal axis, a dot to the right indicates a preference for rightward plans, where a dot to the left indicates a preference for leftward plans, with the radius indicating the magnitude of preference. Why, then, is the 50% mark not at the center of the circle?

This figure has been changed to address this confusion.

Discussion: Line 187-188 reads: “The high degree to which the initial reach plan predicted the free-choice behavior suggests that early responses in PMd strongly influence the eventual movement decision.” This sentence is overly strong. Activity in PMd could derive from a decision being made elsewhere, reflecting (but not driving) a bias toward the initial choice. Please re-phrase this sentence to take this correlation-vs-causation issue into account.

We have changed this wording.

Methods: Line 295-296 reads “We then used the Mahalanobis distance between the neural state and the training set...” I assume that the training set is the set of all single-target reaches described earlier in the paragraph, but this is not explicitly stated.

The methods have been expanded to clarify this point.

In general, the figure text contains many minor errors. Please re-read the figure labels closely and ensure that they are accurate. Below are the issues I found:

Figure 4a: Figure text reads “Blue traces indicate those for which the monkey was cued to the right. Red traces indicate a cue to the right.” I assume one of these colors should correspond to left.

Figure 5: In the figure text, (d) should instead be (c).

Figure 6c: Currently reads “As in a for all reaches to the upper left target...” I assume this should be for the upper right target, based on the figure inset highlighting upper right reaches.

We have made changes to the text and figure captions to address these issues.

Reviewer #3

Major issues and questions

1. The main evidence supporting the claim of a "single reach plan" is the comparison of the covariance in the 1- versus 2-target conditions. Oddly, this analysis is not framed as being the critical test of the single-plan hypothesis, but rather is framed as a validation of being allowed to do dimensionality reduction. The rest of the manuscript after this key result then proceeds to explore the phenomenology of the task (errors, reaction times), with the assumption that the monkey is maintaining only a single movement plan. While these additional observations are interesting, it isn't clear that they provide any additional evidence in favor of a single reach plan.

We acknowledge that the previous submission did not adequately address the question of single v. dual representation. We have added analyses and figures to clearly test these two possibilities. Specifically, we now include simulations of single- and dual-representation (Fig 4) and make direct comparisons with the observed data (Fig 6).

On a related note, I am not convinced that the 'fixed correlation' requirement for dimensionality reduction is really applicable. line 59 - "For this dimensionality reduction approach to be valid, the correlations between neurons must be fixed". I don't follow why this is the case. In general, it doesn't seem to me that individual neurons need to have a fixed pairwise correlation for dimensionality reduction to be valid. Typically, pairs of neurons would have a fixed correlation during repeated movement to the same target, but would be differently correlated for different reach directions. Applying the authors' logic, one should then be prohibited from doing dimensionality reduction for tasks with multiple targets, which is of course absurd. In the example provided by the authors, it seems perfectly valid to employ a dimensionality reduction in which $d1 = (\text{neuron1} + \text{neuron2})$ and $d2 = \text{neuron3}$. I fail to see what the problem would be with this.

We have replaced this analysis and figure.

2. Regarding the hypothesis itself - The authors neglect to mention that several authors have argued in the past that only a single movement plan exists. For instance, Haith et al. (2015) and Wong and Haith (2017) argued on the basis of behavioral evidence that the motor system only ever maintains a single movement plan. Also, Churchland et al. (Neuron, 2010; Nature 2012) proposed that the movement being prepared is determined solely by the initial state of the motor cortex (primarily PMd) at the time of movement initiation. Implicit in this theory is that it is not possible for the motor cortex to simultaneously represent two or more distinct movement plans.

We have incorporated these references. We have also included a brief discussion regarding the implications of dual representation on the dynamical systems framework.

3. The decoding algorithm used to estimate the likelihood is unprincipled and inadequately described. $P(i|D_M(S, \{C_i\}))$, in the equation on line 298, looks like it is supposed to represent a probability (not a likelihood, as claimed in the text - technically the likelihood of target i would simply be $P(S|i)$). However, the term computed is not really a probability - the "probabilities" do not add to one. Properly turning this into a probability would involve applying Bayes' rule (taking $P(i)$ into account). The LHS of line 298 doesn't really correspond to any readily interpretable quantity and, in the end, it feels like an ad hoc metric that seemed to work, rather than the product of a principled approach to inferring the reach plan from the neural data.

We agree that framing it as a likelihood or probability was confusing, and now term it a "Proximity" metric. In subsequent responses, we address questions regarding its formulation.

It is not clear what is meant by $P(D_M(S, \{C_i\})|i)$. How exactly is a probability computed based on the distance? I presume the idea is that $P(S|i)$ has a multivariate normal distribution, in which case $P(S|i)$ depends on the Mahalanobis distance (it would be proportional to $\exp(-D_M(S, \{C_i\})^2)$). Note that the use of the Mahalanobis

distance is actually redundant with conditioning on i . Computed properly, $P(S|i)$ should already involve computation of the Mahalanobis distance.

We calculate probabilities from empirical distributions of the Mahalanobis distance in the 1-Target case. Conditioning on i is redundant, but we include it to highlight the case in which we condition on j (discussed in the following response).

It is not at all clear what the denominator, $P(D_M(S, \{C_i\})|j)$ corresponds to. What does it even mean to condition on j in this case? Does this mean the probability of observing that Mahalanobis distance under the generative model for target j ? If so then why the Mahalanobis distance for i ? What is the rationale for doing this normalization at all? Why not just use max likelihood estimation?

We included the conditioning on j in the denominator (where $j \neq i$) to act as a normalization across reach directions. The clusters associated with the different reach directions were not exactly normally distributed, nor were they equally separable from the other clusters. As a result, distributions of distances were not consistent across directions. This made comparisons between reach directions using raw Mahalanobis distances (or equivalently, probabilities) difficult and potentially misleading. Our metric can be interpreted as: the probability that the neural state reflects a reach plan to target i based only on its distance to cluster i . The denominator serves to marginalize the distance over all reach directions. Note that our conditioning only on the distance to cluster i necessitated the use of empirical distributions, as there is no simple formula by which to calculate the distances from one multivariate normal distribution to another.

We agree that the rationale for this approach was not clearly demonstrated in the initial submission. We have since included analyses and figures showing that the metric is able to discriminate between single- and dual-target representations. We have also included supplementary analysis comparing the Proximity metric described here with traditional classification.

It's not clear why it's necessary to perform dimensionality reduction down to 10 dimensions before doing this analysis. The Mahalanobis distance would naturally ignore (or give a very low weight to) the dimensions with low variance that are discarded during the PCA.

We found calculations in the lower dimensional space to be significantly less noisy compared to the full neural space.

4. An important contribution of the paper is to explain how previous studies could have come to a spurious conclusion of parallel planning. The supplemental materials therefore constitute a critical component of the paper (in my view, much more critical than the bulk of what appears in the main paper). However, these analyses are not explained in sufficient detail. It is not possible to follow the details without referring to the Cisek and Kalaska paper. In particular, the key measure, plotted on the x-axis of Fig S1c,d: "% of PD+OD activity that is greater than the median of orthogonal activity" is not explained at all. Even "OD" is not defined anywhere. Given the importance of this argument to the overall claim, I feel substantially more care is warranted to make this analysis and argument clearer and self-contained (i.e. not needing to refer to Cisek/Kalaska to follow it completely).

We have incorporated the analyses a la Cisek and Kalaska[1] into the main text (Fig 2) along with a full description of the methods.

Regarding the outcome of the analysis, if I have read Figure S2d correctly, it seems to suggest that the existence of a target preference can lead to a conclusion of parallel planning even after controlling for target preferences in the 2-target condition (the distribution for the 60% target preference is shifted above 50%). How can this be reconciled with Figure S1d, which shows that, for the actual data, no such shift happens? Does this imply that the stay-or-switch model used in the simulation isn't an accurate model of what really happened in the task?

We have since reformulated these simulations. The shape of the tuning curves used in the previous simulations introduced a small rightward bias in the distribution even on 1-Target trials. We have since switched from using wholly artificial neurons and now base our simulations on actual neural responses.

Additional Comments:

Was there any relationship between plan "strength" and movement speed? If so, this might suggest that both RT and tuning strength might both be consequences of the monkey's motivational state, rather than the RT being a consequence of the preparatory state.

There is a correlation between RT and peak speed, so it is difficult to disentangle these two possibilities. While we suggest something of a causal relationship between preparatory state and movement properties (e.g., RT, speed), it is possible that both are simply responses to a third factor (like motivation). We have included this possibility in the discussion.

What is the rationale for taking the square root of firing rate?

The square root transform is commonly used[2-4] as a soft normalization of firing rates across neurons. It is especially useful for PCA, as it prevents a few neurons with high firing rates from dominating the total variance.

hyphenation errors throughout:

line 31 ("a multiple-potential-target reaching task")

line 121 ("single reach plans")

line 166 ("two-target tasks")

line 230 ("target-on period")

line 232 ("target-off period")

line 240 ("One-target trials")

line 261 ("the desired-reach-direction-tuning-related covariance") - This one is a bit much even when correctly hyphenated. might be worth rephrasing.

line 294 ("post-target-appearance")

line 234 - "...the correct TARGET".

We have made a number of changes to the text that address these issues.

References

1. Cisek, P. and J.F. Kalaska, *Neural correlates of reaching decisions in dorsal premotor cortex: specification of multiple direction choices and final selection of action*. *Neuron*, 2005. **45**(5): p. 801-14.
2. Churchland, M.M., et al., *Stimulus onset quenches neural variability: a widespread cortical phenomenon*. *Nat Neurosci*, 2010. **13**(3): p. 369-78.
3. Rouse, A.G. and M.H. Schieber, *Spatiotemporal Distribution of Location and Object Effects in Primary Motor Cortex Neurons during Reach-to-Grasp*. *J Neurosci*, 2016. **36**(41): p. 10640-10653.
4. Byron, M.Y., et al. *Gaussian-process factor analysis for low-dimensional single-trial analysis of neural population activity*. in *Advances in neural information processing systems*. 2009.

REVIEWERS' COMMENTS:

Reviewer #1 (Remarks to the Author):

Generally I think the manuscript improved a lot. I have the following comments:

- Figure 1b should show a scale, it might even be helpful to show a larger portion of the brain here to make it easier to compare.**
- p7 In the text "The analysis in Figure 2c was designed..." should that not be Fig. 2d?**
- Fig. 7a in the rebuttal the authors stated that they realigned the traces in the blank period to the cue. This should be made visually clear and also the alignment should be stated in the figure caption for all epochs.**

- I added some additional comments to the rebuttal which I attached as a separate file.**

[Editorial Note: See the following page for Reviewer #1's additional comments.]

Response to Reviewers

We wish to thank the reviewers for their many insightful comments. We have attempted to address all that were feasible. Unfortunately, as neither monkey is still available, we are unable to add any more data to the study.

Reviewer #1

Major issues and questions

1. Even though the authors did only record from a relatively small part of PMd they make a very strong claim in the abstract and conclusion of the manuscript: “Our results suggest that cortex plans only one option at a time...”. One of the studies they cited, Cisek & Kalaska 2005, reported a caudal-rostral gradient of target specificity. More rostral areas would contain more cells that are active for potential reach goals. Could it be that the array placement was too close to M1 to yield similar results as reported previously? I find it a bit hasty to conclude the cortex would only ever plan one option at a time considering that there have been studies in different parts of premotor cortex and parietal including a whole theory about how more vague plans are transformed to more concrete plans the closer to motor one gets.

We acknowledge that our findings cannot (and should not) be extrapolated to all of cortex, and we have reflected that limitation in the text. However, based on the array placements (Fig 1) and prominence of planning-related activity in the averaged responses (Fig 2) we are reasonably confident that the recordings were made from dorsal premotor cortex. We did not observe any difference in response characteristics between the rostral and caudal halves of the array, likely due to the small region of cortex it covered.

I do not doubt that you recorded from PMd, the question was if you might have recorded primarily from the caudal part. In their C & K 2005 paper the authors do report that they found no or only few neurons with represented dual targets (called potential response cells in the paper) in the caudal region of PMd (F2). The representation of potential targets seem to lie further rostral (F7). I tried to compare the array locations depicted in your Fig. 1 (which I found relatively hard since there are no scales in Fig 1b, but since the array is 4x4 mm I had a rough idea) with Fig. 5 in the C & K paper and to me it looks as if you were pretty much at the caudal tip for monkey C and maybe closer to the relevant part for monkey M. If that is the case then this could be an explanation for your results, but it would also not really answer the question of multiple representations. As far as I know nobody doubts that at the end of the processing stream in the motor cortex (or very close to it) there is only one representation

In Figure 2 you demonstrate a weakness in the original argument from the C & K paper which shows that you could get a result that looks as if it was a two target representation but actually is not. This is a good demonstration, however if you were in a cortical region further downstream where initially multiple reach plans collapse to a single reach plan you would come to the same conclusion. I think it would greatly benefit the paper if you could also provide a plot similar to Figure 5b from the C & K paper (instead of the rostral caudal gradient you could show this for each of the sessions/monkeys) using the data you already have. If you also find a lot of “potential response” cells this would clearly show that your results are not an effect of being too close to motor (caudal PMd).

2. I think the analysis of the neural state is a very interesting and important point of the paper. Still for comparison sake it would have been useful to also provide a more in-depth analysis of single cell selectivity. In the supplementary material one single cell analysis is performed but it would have been more informative to show single cell behavior in more detail.

We have included trial-averaged results of single-neuron activity in a new figure (Fig 2). However, due to space constraints, we have chosen not to show single-neuron rasters, tuning curves, etc. As discussed in the text, the trial-averaging required to describe single-neuron responses may lead to misleading results. We believe that the state-space analyses (which have been significantly clarified) are most appropriate for addressing the main question of single- or dual-representation.

I see your point and I think the new figure 2 helps to explain things better. Since your paper heavily depends on comparison to the C & K paper and my comment 1 I still think some population statistics are necessary.

3. Interestingly even though they recorded in PMd and M1 results from M1 seem to be completely omitted in the manuscript. Why was the same analysis not applied to the M1 data that they must have recorded? This would have helped to compare their results better to previous studies that they cited.

We have now included M1 results in the supplementary section (Fig S1). M1 exhibited less planning-related activity, but the state-space analysis results were almost identical to those from PMd.

I think this is very helpful. Since the results are so similar this could actually support the too-close-to-motor hypothesis from my comment 1.

4. One of the main points of the paper is shown in figure 4a. Here the authors argue that 88% of all trajectories are biased for a movement in one direction during the blank period. Looking at the start of the cue period it looks as if this ratio must have changed near the end of the blank period. Since the blank period was variable in length it would have been interesting to see how the trajectories would look like if they were aligned to the cue and then tracing them backwards to understand what happened at the end of the blank period. Even in those traces they show in the blank period it looks as if a shift happens towards 0 likelihood. What is the explanation for this? Also, why not show a plot like 4a for all directional pairs and all sessions?

We have realigned the traces to the Cue to make trends more apparent. The tendency of the decoded plan strength to drift towards 0 after target disappearance is not consistent across sessions. It is therefore difficult to determine whether the apparent “weakening” is a noteworthy effect. To address the point that we showed only one target pair, we now include plots that incorporate data from all trials and all sessions.

The alignment should be stated somewhere and/or should be made more visible. In the figure caption for figure 7 (the new figure 4 I presume) still states that the data is from one session. The ratio is still 88 %/ 12 % sounds strange if this is combined data now. Also it does not say from which monkey.

5. The authors mention that in other experiments it has been tried to unbiase the monkeys. Has there been any attempt to try this and see if the neural state trajectories would change then?

Unfortunately, the monkeys in this study are no longer available. However, we included as a discussion point the effect “unbiasing” might have on planning responses. We also discuss the implications of training, and whether the subsequent development of a neural effect would reflect natural sensorimotor control or something like specialized skill learning.

6. This paper heavily depends on comparisons to Cisek & Kalaska 2005 (and others). If the main point was to reevaluate the those results why hasn't the task paradigm be exactly replicated? One of the problems I see here is for example that the important blank period is kept rather short in this experiment (250-500 ms vs. more than 1000 ms in other cited papers).

The simplicity in task design allowed us to evaluate “natural” planning behavior on a two-target task while limiting potential effects of overtraining. We intended to later transition to the exact version used by Cisek and Kalaska (2005), but were unsuccessful due to monkey behavioral issues and then array failure. The similarity in trial-averaged activity between our results and those from Cisek and Kalaska (2005) suggest that our task did induce similar neural responses. It seems likely that the major results reported in the literature (e.g., dual representation) would be consistent across task variants.

7. Since the strategies the monkeys use and the biases for specific directions depends a lot on the training history it would be interesting to know how the training was done and how many training sessions had been done before the analysis phase.

The monkeys were already trained on a simple 1-Target center-out task. For the present study, they were shown the 2-Target task as described until they could perform at an above-chance level. This took about 3 sessions for Monkey C and 2 sessions for monkey M. We did not find the directional preferences to be consistent or follow a clear pattern. We now also comment on the potential effect of training on the representation of multiple reach targets in the discussion section.

8. Since the claim is made that the cortex plans only one reach option at a time how are those trajectories interpreted which are close to 0 likelihood? There are clearly some cases when the bias was only very weak or not present at all.

We have included new figures (Figs 3,5) that should help address this question. In brief, decoded magnitudes near 0 should be interpreted as “no reach plan” rather than “two opposing and equally strong reach plans”.

Ok, how would the plots look like if there was a dual representation?

9. What was the total number of recording sessions and trials per session? It sounds as if there were only three sessions which is rather low especially considering that there might be training effects which could change the strategies the monkeys where using and the neuronal representation.

Each monkey completed 3 full sessions (after training) of the task as described. Sessions averaged 400 trials (min: 227, max: 560). While the session count was a bit low, the amount and quality of recorded neurons, and the consistency across sessions and monkey appear to be sufficient to answer the questions posed. We include the question of trained neuronal representations as a discussion point.

Reviewer #2 (Remarks to the Author):

Dekleva and coauthors have significantly revised and improved this manuscript from the previous submission. In particular, they added several figures showing how neural activity on individual 2-target trials compares to neural activity for non-ambiguous reaches to the same targets. They convincingly show that the predictions of a "simultaneous plan" model would show very different neural behavior than they observed in their own data. In particular, their data tended to display "low" proximity to one target, and "high" proximity to the other (similar to trials in which only one target was present), whereas simultaneous models ought to have proximity somewhere in the middle for both.

The addition of further analyses to discriminate between the simultaneous and single-plan hypotheses in figures 2-5 were well-done and important to the conclusions of the paper. I am happy to recommend this paper for publication with no further major revisions; I have appended a short list of small edits for clarity.

Lines 51-52: In the task description, you state the targets "disappeared for 250-500 milliseconds (Target Blank) before reappearing for another 250-500 milliseconds (Cue)." This makes it sound as though both targets reappeared, rather than just one, as a disambiguation.

Line 139: "we next applied it to activity on 2-Target trials" (The to is missing)

Line 143: "mechanisms by which PMd could simultaneously represent two reach directions" (the ly is missing).

Reviewer #3 (Remarks to the Author):

The authors have extensively revised the paper, much for the better. They developed a far more comprehensive analysis that significantly improves the paper and rightly focuses the manuscript on the core question of single versus multiple movement plans. I find this revised article well written and convincing.

I have only a few minor outstanding comments:

I don't quite follow the cosine-tuning aspect of the simulations (described in lines 177-185). Why was this necessary? Would it not be possible to instead construct a surrogate dataset from activity in single-target trials (as in the Fig. 2 analysis).

line 548 - the definition of reaction time in terms of movement along a specific axis could be problematic - high RTs might reflect transiently moving in the wrong direction. It's unclear whether the velocity alternative is similarly defined with respect to a specific direction. Defining RT based on hand speed (i.e. regardless of direction) would be better.

line 186 - 'averaged-target' model (line 186) could be slightly misleading. At first I thought that meant the expected neural activity for the average target location, which wouldn't be a crazy idea if the targets weren't spaced 180 degrees apart. 'averaged representation' may be more accurate.

218 - would be helpful to define the DRI more concretely here (or at least point to the

methods, as it is not that straightforward). Even if the methods, I found there was little explanation how this measure was arrived at. I'm okay with it being somewhat ad hoc, but a little more explanation would be nice (e.g. why multiply by the mean-squared proximity?)

line 472-473 - "We then calculated preferred axes by doubling the angles in the cosine model and fitting to 2-Target activity in the same window" I didn't find this clear in the context. What is a preferred axis vs a preferred direction? Which angles (PD angle) and what is the point of doubling them? Please clarify.

line 473-475 unclear what "magnitude of the cosine model fit" means here. Also, 'significance' is vague. Please be more specific and concrete here as some readers will be confused.

Errors/Typos:

line 61: hyphenation: multiple-potential-target reaching task

line 170: applied it TO activity

line 174: simultaneousLY

line 199 - reference to open versus closed circles, when they are actually black and gray.

line 464 - cosign -> cosine

Reviewer #1

Generally I think the manuscript improved a lot. I have the following comments:

1. Figure 1b should show a scale, it might even be helpful to show a larger portion of the brain here to make it easier to compare.

We have added a scale bar to the figure per request, as well as a label indicating anatomical directions. However, we are only able to faithfully report landmarks that were exposed by the craniotomy. We do believe that the visible features, combined with additional analyses (see Supplementary Figure 2), should be adequate to place the arrays on the rostral/caudal divide of PMd.

2. p7 In the text "The analysis in Figure 2c was designed..." should that not be Fig. 2d?

We have changed the wording to clarify that the control analysis as used in Fig. 2c (initially described by Cisek and Kalaska [2005]) implicitly tests for the presence of an unbiased guessing approach. We then apply the analysis to artificial datasets of unbiased guessing (Fig 2d) and biased guessing (Fig 2e) and report that the test can successfully identify single reach plans in the former but not the latter.

3. Fig. 7a in the rebuttal the authors stated that they realigned the traces in the blank period to the cue. This should be made visually clear and also the alignment should be stated in the figure caption for all epochs.

We now clearly state in the figure caption the alignments used throughout the trial. The traces during Target Blank are right-aligned (to Cue).

4. I do not doubt that you recorded from PMd, the question was if you might have recorded primarily from the caudal part. In their C & K 2005 paper the authors do report that they found no or only few neurons with represented dual targets (called potential response cells in the paper) in the caudal region of PMd (F2). The representation of potential targets seem to lie further rostral (F7). I tried to compare the array locations depicted in your Fig. 1 (which I found relatively hard since there are no scales in Fig 1b, but since the array is 4x4 mm I had a rough idea) with Fig. 5 in the C & K paper and to me it looks as if you were pretty much at the caudal tip for monkey C and maybe closer to the relevant part for monkey M. If that is the case then this could be an explanation for your results, but it would also not really answer the question of multiple representations. As far as I know nobody doubts that at the end of the processing stream in the motor cortex (or very close to it) there is only one representation.

In Figure 2 you demonstrate a weakness in the original argument from the C & K paper which shows that you could get a result that looks as if it was a two target representation but actually is not. This is a good demonstration, however if you were in a cortical region further downstream where initially multiple reach plans collapse to a single reach plan you would come to the same conclusion. I think It would greatly benefit the paper if you could also provide a plot similar to Figure 5b from the C & K paper (instead of the rostral caudal gradient you could show this for each of the sessions/monkeys) using the data you already have. If you also find a lot of "potential response" cells this would clearly show that your results are not an effect of being too close to motor (caudal PMd).

We thank the reviewer for this suggestion, and have added Supplementary Figure 2 to address the concern of array placement and response characteristics. For this analysis, we categorized the single neuron responses according to three groups (Movement, Selected, and Potential) in a manner similar to Cisek and Kalaska (2005). The most important neurons with respect to PMd identification were the Potential neurons, which appeared bimodally-tuned on trial-averaged data of the Target On epoch (2-Target). The prevalence of

Potential neurons in each array region matched those expected based on the anatomical locations of the arrays. The PMd array for monkey M was placed more rostrally, and had an equally high proportion of Potential neurons across its anterior and posterior halves. The PMd array for monkey C was placed closer to M1. Its posterior half contained a somewhat greater proportion of Potential neurons than the M1 array. Neurons in the anterior half of the array displayed predominantly non-motor-like activity, with a high incidence of Potential neurons.

The higher incidence of Potential neurons in PMd (compared to M1) for both monkeys suggests that arrays were placed in sufficiently anterior brain regions. However, we wanted to take additional measures to ensure that our population results did not simply reflect activity from more motor-like cells. We repeated the major population analyses (from Figures 5 and 6) on only Potential neurons identified in PMd. This subset of neurons was the least motor-like in their responses, as they exhibited early, apparent bimodal tuning on 2-Target trials. However, we found the same qualitative results as for the whole population, with no evidence of dual representation (Supplementary Figure 2c-f).

To further illustrate that the neurons recorded from PMd were not indistinguishable from those in M1, we have added the control analysis from Cisek and Kalaska (as in Fig 2) for M1 in Supplementary Figure 1b. The histograms of high PD and anti-PD activity are both centered at 50%, indicating no presence of dual representation. This is in stark contrast to the highly right-skewed distribution observed for PMd in Figure 2b. Thus, both trial-averaged metrics used (neural response type and the control from Cisek and Kalaska) revealed marked differences between M1 and PMd.

5. We have included trial-averaged results of single-neuron activity in a new figure (Fig 2). However, due to space constraints, we have chosen not to show single-neuron rasters, tuning curves, etc. As discussed in the text, the trial-averaging required to describe single-neuron responses may lead to misleading results. We believe that the state-space analyses (which have been significantly clarified) are most appropriate for addressing the main question of single- or dual-representation.

I see your point and I think the new figure 2 helps to explain things better. Since your paper heavily depends on comparison to the C & K paper and my comment [4] I still think some population statistics are necessary.

Our response to the previous comment outlines the addition of single-neuron population statistics.

6. We have now included M1 results in the supplementary section (Fig S1). M1 exhibited less planning-related activity, but the state-space analysis results were almost identical to those from PMd.

I think this is very helpful. Since the results are so similar this could actually support the too-close-to-motor hypothesis from my comment [4].

In addition to the following response, see point 4 above. The population neural state results were similar between M1 and PMd in that both appeared to contain only one reach plan at a time and the directions of those decoded reaches agreed with each other. However, the activity from the two areas was clearly distinguishable, as evidenced by the guessing control analysis (compare PMd in Figure 2c to M1 in Supplementary Figure 1b) and neural response types (compare PMd to M1 in Supplementary Figure 2a,b).

7. We have realigned the traces to the Cue to make trends more apparent. The tendency of the decoded plan strength to drift towards 0 after target disappearance is not consistent across sessions. It is therefore difficult to determine whether the apparent “weakening” is a noteworthy effect. To address the point that we showed only one target pair, we now include plots that incorporate data from all trials and all sessions.

The alignment should be stated somewhere and/or should be made more visible. In the figure caption for figure 7 (the new figure 4 I presume) still states that the data is from one session. The ratio is still 88 %/ 12 % sounds strange if this is combined data now. Also it does not say from which monkey.

See response to comment 3. The data in Figure 7a is indeed from the same example session (monkey now noted in the caption) as before. However, Figures 3, 5, and 6 include all data from all sessions. We did not find it feasible (or informative) to include temporal traces from every target axis, session, and monkey.

8. We have included new figures (Figs 3,5) that should help address this question. In brief, decoded magnitudes near 0 should be interpreted as “no reach plan” rather than “two opposing and equally strong reach plans”.

Ok, how would the plots look like if there was a dual representation?

If there was dual representation, the temporal traces of Δ Proximity would indeed lie close to 0. However, we show in Figures 4-5, and especially Figure 6, that the absolute Proximity values contain no evidence of dual representation. Thus, all subsequent Δ Proximity values near zero should be interpreted as reflecting weak-to-nonexistent reach plans.

Reviewer #2

Dekleva and coauthors have significantly revised and improved this manuscript from the previous submission. In particular, they added several figures showing how neural activity on individual 2-target trials compares to neural activity for non-ambiguous reaches to the same targets. They convincingly show that the predictions of a “simultaneous plan” model would show very different neural behavior than they observed in their own data. In particular, their data tended to display “low” proximity to one target, and “high” proximity to the other (similar to trials in which only one target was present), whereas simultaneous models ought to have proximity somewhere in the middle for both.

The addition of further analyses to discriminate between the simultaneous and single-plan hypotheses in figures 2-5 were well-done and important to the conclusions of the paper. I am happy to recommend this paper for publication with no further major revisions; I have appended a short list of small edits for clarity.

1. Lines 51-52: In the task description, you state the targets “disappeared for 250-500 milliseconds (Target Blank) before reappearing for another 250-500 milliseconds (Cue).” This makes it sound as though both targets reappeared, rather than just one, as a disambiguation.

We have clarified this wording.

2. Line 139: “we next applied it to activity on 2-Target trials” (The to is missing)

Fixed

3. Line 143: “mechanisms by which PMd could simultaneously represent two reach directions” (the ly is missing).

Fixed

Reviewer #3

The authors have extensively revised the paper, much for the better. They developed a far more comprehensive analysis that significantly improves the paper and rightly focuses the manuscript on the core question of single versus multiple movement plans. I find this revised article well written and convincing.

I have only a few minor outstanding comments:

1. I don't quite follow the cosine-tuning aspect of the simulations (described in lines 177-185). Why was this necessary? Would it not be possible to instead construct a surrogate dataset from activity in single-target trials (as in the Fig. 2 analysis).

For Figure 2, we simulated strategies that employed *single* reach plans. Thus, simply relabeling 1-Target activity (which we can safely assume contained only single plans) was sufficient for obtaining perfect simulations. However, it is not possible to similarly construct a dual-representation dataset from 1-Target activity without some model of 2-Target neural responses. We presented two different models: cosine tuning (with doubled frequency), and a nonparametric averaged-plan model. The doubled cosine model assumes (as reported by Cisek and Kalaska) that a neuron will increase its firing in response to the presence of either of two targets. The averaged-plan model assumes that a neuron's response to the presence of two targets will be the average of its response to each one individually. There are many other possible models of dual representation possible, but we thought these two represented the most likely candidates based on previous reports.

2. line 548 - the definition of reaction time in terms of movement along a specific axis could be problematic - high RTs might reflect transiently moving in the wrong direction. It's unclear whether the velocity alternative is similarly defined with respect to a specific direction. Defining RT based on hand speed (i.e. regardless of direction) would be better.

We agree that our initial method for determining reaction time may have been susceptible to the issues stated. We have therefore rerun the analysis using a hand speed threshold (>5 cm/s) to determine reaction time. The results were not qualitatively affected by this alteration.

3. line 186 - 'averaged-target' model (line 186) could be slightly misleading. At first I thought that meant the expected neural activity for the average target location, which wouldn't be a crazy idea if the targets weren't spaced 180 degrees apart. 'averaged representation' may be more accurate.

We agree that the term could be misleading for non-opposed target cases. We now refer to it as the "averaged-plan" model, which we believe is less likely to cause the same confusion.

4. 218 - would be helpful to define the DRI more concretely here (or at least point to the methods, as it is not that straightforward). Even if the methods, I found there was little explanation how this measure was arrived at. I'm okay with it being somewhat ad hoc, but a little more explanation would be nice (e.g. why multiply by the mean-squared proximity?)

We have added additional explanation in the methods for the rationale of the metric. In short, the metric multiplies two separate values. The first term scales by the distance from the origin and the second scales by the angle from the diagonal. By combining these two terms, the final metric returns low values for points along either axis or the origin (as would be seen for single reach plans) and high values for points along the diagonal but away from the origin (as would be seen for dual reach plans). We note that the final metric is actually quite similar to the much simpler $\min(P(S,a), P(S,b))$. However, the DRI metric as calculated (1) is more generous to the dual plan hypothesis, as it rewards points that lie near the diagonal, and (2) provides better separation between simulated dual- and single-reach plans.

5. line 472-473 - "We then calculated preferred axes by doubling the angles in the cosine model and fitting to 2-Target activity in the same window" I didn't find this clear in the context. What is a preferred axis vs a preferred direction? Which angles (PD angle) and what is the point of doubling them? Please clarify.

We have adjusted the wording in that section to clarify. Briefly, fitting a cosine model after doubling the angles returns a bimodal function over the range $[-\pi, \pi]$. This bimodal function is the dual-representation equivalent of the standard cosine tuning curve (all opposing directions are equal). The two peaks of the curve define the preferred axis.

6. line 473-475 unclear what "magnitude of the cosine model fit" means here. Also, 'significance' is vague. Please be more specific and concrete here as some readers will be confused.

We have changed the wording, and now refer to the "amplitude term" of the cosine fit. The meaning of significance is clarified in the subsequent sentence by describing the cutoff criterion from the bootstrap procedure.

7. Errors/Typos:

line 61: hyphenation: multiple-potential-target reaching task

line 170: applied it TO activity

line 174: simultaneousLY

line 199 - reference to open versus closed circles, when they are actually black and gray.

line 464 - cosign -> cosine

We have addressed all of these errors, with the exception of our references to open and closed circles. Those descriptors are used in the text to describe Figure 5 (open/filled blue and red circles), not Figure 4 (black and gray).